# STAT1-IFITM3 promotes autophagy in epithelial cells to control *Cryptosporidium parvum* infection

Lijun Cui[1,2,3], Teng Li[1,2,3], Jing Zhang[1,2,3], Yujuan Shen[1,2,3], Jianping Cao[1,2,3,4]

***Cryptosporidium parvum*** **infects intestinal epithelial cells, causing cryptosporidiosis in humans and neonatal livestock. Autophagy is an important component of the defense against *C. parvum* infection. IFITM3 is an IFN-stimulated gene that regulates viral infection through autophagy; however, its role in *C. parvum* infection is unknown. Herein, IFITM3 levels increased significantly after *C. parvum* infection and promoted autophagy. *IFITM3* overexpression up-regulated LC3B II/LC3B I, Beclin-1, ATG7, and ATG5, but down-regulated p62 during infection. In *C. parvum*-infected HCT-8 cells, IFITM3 interacted with LC3B, Beclin-1, and ATG5. *IFITM3* knockdown or autophagy-inhibitor treatment decreased the ratio of LC3B II/LC3B I, Beclin-1, ATG7, and ATG5, but increased p62 expression. Furthermore, IFITM3-regulated autophagy was associated with the inflammatory response, cell survival, and *C. parvum* clearance. *IFITM3* overexpression inhibited apoptosis, increased inflammatory cytokine IL-8 production, and decreased the cellular *C. parvum* burden. *IFITM3* silencing had the opposite effects. IFITM3 expression was positively regulated by STAT1 during infection, whereas *IFITM3* knockdown increased STAT1's compensatory effect. Thus, intestinal epithelial cells resist *C. parvum* infection via autophagy through the STAT1-IFITM3 axis.**

## Introduction

*Cryptosporidium* is an important opportunistic pathogenic protozoa that causes gastroenteritis in a variety of vertebrate hosts, including humans (Bouzid et al, 2013). Acute gastrointestinal symptoms can be prolonged but are usually self-limiting (Checkley et al, 2015). Cryptosporidiosis can be more serious in immuno-compromised individuals, and in those who are malnourished, have chronic diarrhea, and those with hepato-biliary tree and extra-gastrointestinal site infection; however, few options for treatment or prevention exist (Chalmers et al, 2019). Currently, cryptosporidiosis is increasingly identified as an important cause of morbidity and mortality worldwide.

As an intracellular parasite, *Cryptosporidium parvum* oocysts can attach to the surface and enter intestinal epithelial cells (IECs), forming extracytoplasmic vacuoles under the cell membrane, and using the host nutrients for reproduction (O'Hara et al, 2005; Tandel et al, 2019). Studies have shown that *C. parvum* infection can significantly up-regulate the expression of Toll-like receptors (TLR2 and TLR4) in IECs and activate NF-κB signaling pathways (Yang et al, 2015; Zhang et al, 2020). Then, *C. parvum* increases the secretion of TNF-$\alpha$ and IL-8, and further initiates the defense response of IECs and other innate immune cells (McDonald et al, 2013; Ch Stratakos et al, 2017; Rahman et al, 2022). IFN-$\beta$ has opposite effects, mainly inhibiting *cryptosporidium* growth, but also promoting *cryptosporidium* growth in some contexts (Deng et al, 2023). Host cell-intrinsic recognition of *C. parvum* results in IFN-$\lambda$ production, which promotes IEC defense against *C. parvum* by inhibiting parasite invasion and mitigating the loss of the paracellular barrier function (Ferguson et al, 2019; Gibson et al, 2022). Studies have also found that the up-regulated expression of IFN-$\alpha$ and IFN-$\gamma$ can directly inhibit the development process of *C. parvum* after infection (Barakat et al, 2009; Gullicksrud et al, 2022). In addition, IECs can also up-regulate the secretion of IL-15 and regulate the accumulation of inflammatory sites of natural killer (NK) cells to jointly eliminate *C. parvum* (Dann et al, 2005). Therefore, as the main host cells of *C. parvum* infection, IECs play a core role in activating and coordinating the host immune response during anti-infection immunity.

Autophagy is an evolutionarily conserved catabolic cycle pathway, in which long-lived cellular proteins and some dysfunctional organelles are degraded by lysosomes to maintain cellular homeostasis (Glick et al, 2010; Saha et al, 2018). Autophagy involves more than 40 proteins and is regulated by nutrient availability and various stress sensing signaling pathways (Saha et al, 2018). In the intestinal mucosa, autophagy has been shown to be important in maintaining the integrity of the anti-microbial defense epithelial barrier and mucosal immune response (Patel & Stappenbeck, 2013; Haq et al, 2019). Autophagy plays an important role in both innate and adaptive immune defense mechanisms against viral, bacterial,

[1]National Key Laboratory of Intelligent Tracking and Forecasting for Infectious Diseases, National Institute of Parasitic Diseases at Chinese Center for Disease Control and Prevention, Chinese Center for Tropical Diseases Research, Shanghai, China    [2]Key Laboratory of Parasite and Vector Biology, National Health Commission of the People's Republic of China, Shanghai, China    [3]World Health Organization Collaborating Centre for Tropical Diseases, Shanghai, China    [4]The School of Global Health, Chinese Center for Tropical Diseases Research, Shanghai Jiao Tong University School of Medicine, Shanghai, China

Correspondence: caojp@chinacdc.cn

and parasitic infections (Deretic et al, 2013). On the one hand, autophagy can mediate the clearance of intracellular pathogens, such as *Streptococcus*, *Salmonella*, *Shigella*, and *Listeria*, by targeted degradation in IEC lysosomes, thereby eliminating the pathogens and protecting the intestinal epithelial barrier (Jo et al, 2013; Hu et al, 2020). On the other hand, pathogens might induce or inhibit the autophagy of host IECs to promote their own survival, proliferation, and infectivity in host cells (Escoll et al, 2016; Xiong et al, 2019). For example, the diarrheal pathogen *Clostridium difficile* has been shown to induce a strong autophagy response in IECs in vitro to enhance the disease-causing processes (He et al, 2017). Similarly, some protozoan parasites can manipulate host autophagy to evade host clearance (Onizuka et al, 2017; Portillo et al, 2017). It has been confirmed that *C. parvum* infection induces autophagy in IECs. Modulation of host cell autophagy by *C. parvum* could result in alterations to pivotal host cell processes, for example, expression of epithelial junctional proteins, which might be a key cause of *C. parvum* pathophysiology (Priyamvada et al, 2021). A case-control study found that autophagy-related 16-like 1 single nucleotide gene polymorphism increases the risk and severity of *C. parvum* infection (El-Refai et al, 2021). A recent study reported that long non-RNA (lncRNA) *Nostrill* facilitates the transcription of *Igtp*, *Gadd45g*, and *iNos* induced by IFN-γ, which in turn positively regulates autophagy, ultimately contributing to the epithelial cell-intrinsic defense against *Cryptosporidium* infection (Sharmin et al, 2024). *C. parvum* regulates the autophagy of HCT-8 cells (human ileocecal colorectal adenocarcinoma cells (IECs)) to facilitate survival by inhibiting miR-26a and promoting miR-30a expression (Jiang et al, 2022). In addition, *C. parvum* could maintain intracellular survival by inhibiting autophagy via the epidermal growth factor (EGFR)-phosphatidylinositol-4,5-bisphosphate 3-kinase (PI3K)/ protein kinase B (Akt) pathway (Yang et al, 2023). However, the key proteins and signaling pathways in IECs that induce autophagy after *C. parvum* infection, and their role in immunity against *C. parvum*, remain to be further explored.

Interferon-induced transmembrane protein 3 (IFITM3) is a natural immune response protein that can be induced by viral infection and IFN (Jiménez-Munguía et al, 2022). As an interferon-stimulated gene (ISG), it can inhibit the invasion and intracellular replication of various viruses, such as Dengue virus, West Nile virus, flavivirus, and severe acute respiratory syndrome coronavirus, and has important antiviral effects. Individuals lacking IFITM3 are also susceptible to low pathogenic influenza viruses (Brass et al, 2009). Therefore, IFITM3 has immune protection abilities, and is considered to be the first line of cellular defense against viruses. Amplification of the PI3K signaling pathway dependent on IFITM3 partially acts on the downstream pathway of BCR activator of RhoGEF and GTPase (BCR), which is crucial for the rapid amplification of B cells with high affinity for antigens (Lee et al, 2020). The continuous expression of IFITM3 can maintain the long-term survival of lung-resident memory T cells and protect the body from infection when exposed to viruses (Wakim et al, 2013). In mouse neural retinal progenitor cells, IFITM3 plays a significant role in maintaining the homeostasis of progenitor cell self-renewal by sustaining low-level activation of chaperone-mediated autophagy (CMA) to eliminate deleterious factors from cells (Jin et al, 2022). High expression of IFITM3 has been reported to induce autophagy,

including microtubule-associated protein 1 light chain 3 (LC3) puncta and lipidation, during influenza A virus infection, thereby limiting viral replication (Feeley et al, 2011; Yount et al, 2012). In addition, IFITM3 mediates autophagic degradation of interferon regulatory factor 3 (IRF3) and negatively regulates virus-induced hyperintense type I IFN-mediated inflammatory tissue damage (Jiang et al, 2018). Proteomic analysis in our laboratory found that the gene and protein expression levels of IFITM3 in the HCT-8 cell line were significantly increased at 36-h post *C. parvum* infection (Li et al, 2021). Considering the results of the above studies, we hypothesized that high expression of IFITM3 after *C. parvum* infection might participate in the host anti-parasite immune defense process by regulating IEC autophagy to limit the excessive inflammatory response after *C. parvum* infection; thus playing an important role in the host anti-parasite immunity. Therefore, this study aimed to test this hypothesis by investigating the mechanism by which IFITM3 might regulate autophagy after *C. parvum* infection.

## Results

### *C. parvum* infection enhanced the expression of IFITM3 and induced autophagy in HCT-8 cells

Previously, proteomic analysis in our laboratory revealed that the level of IFITM3 in HCT-8 cells increased significantly and the IFN signaling pathway was the most significantly enriched biological process after *C. parvum* infection (Li et al, 2021). qRT-PCR results showed that mRNA levels of *IFITM3*, *IFN-α1*, and *IFN-β* were up-regulated in response to 24 h infection of *C. parvum* (Fig S1A–D). To explore the role of IFITM3 expression during *C. parvum* infection, IFITM3 protein levels in HCT-8 cells were detected using Western blotting. The results showed that the level of IFITM3 increased from 0 to 48 h post *C. parvum* infection (hpi) (Fig 1A and B). Typically, microtubule-associated protein 1A/1B light chain 3B (LC3B) II/LC3B I levels increase with autophagic activity and p62 levels decrease. To confirm whether autophagy occurred in HCT-8 cells infected with *C. parvum*, the levels of LC3B and p62 were measured using Western blotting. The results indicated that LC3B II/LC3B I increased gradually from 0 to 48 h post infection (Fig 1A and C). By contrast, the levels of p62 showed no change from 0 to 8 h and 36 to 48 h but increased at 12–24 h of infection (Fig 1A and D). Moreover, the levels of LC3B II/LC3B I increased gradually whereas p62 was decreased from 0 to 48 h during treatment with an autophagy activator (Rapamycin) (Fig 1E–G). These results indicated that *C. parvum*-induced autophagy in HCT-8 cells; however, autophagy flow was blocked, paused, or it was not induced at 12 and 24 h.

### Knockdown of *IFITM3* decreased autophagy of HCT-8 cells after *C. parvum* infection

IFITM3 plays an important role in autophagy during some viral infections. In this study, to investigate the role of IFITM3 in regulating autophagy to control *C. parvum* infection, a small interfering RNA (siRNA) was used to knock down *IFITM3* in HCT-8 cells. The levels of IFITM3 and LC3B were assessed using Western blotting.

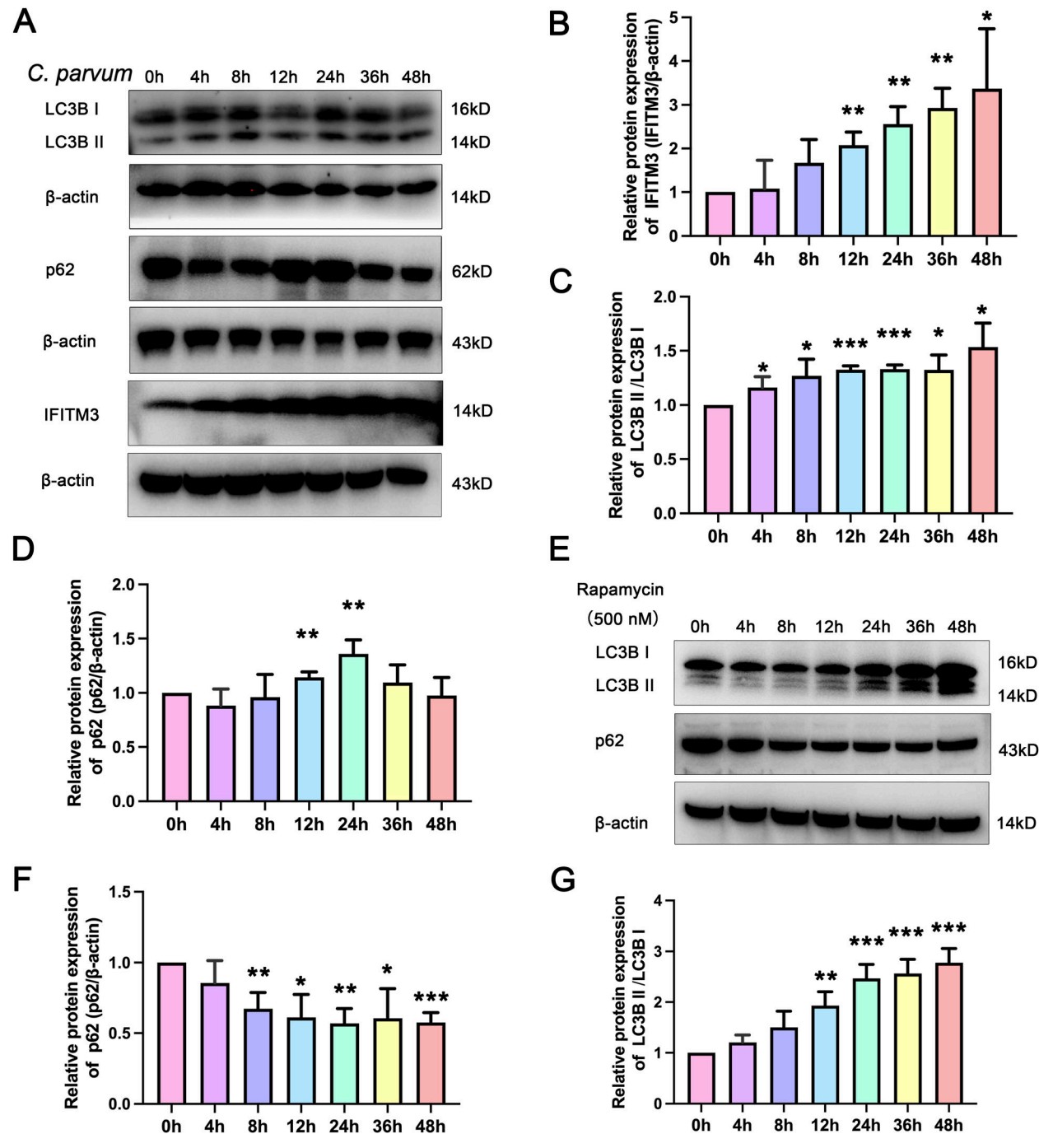

**Figure 1. *Cryptosporidium parvum* infection results in the increased expression of IFITM3 and the occurrence of autophagy in HCT-8 cells.**
**(A)** Western blotting analysis of the levels of IFITM3, LC3B, and p62 from 0 to 48 h post infection. **(B, C, D)** Grayscale analysis of the IFITM3 (B), LC3B (C), and p62 (D) Western blotting results. **(E)** Western blotting analysis of the expression levels of LC3B and p62 from 0 h to 48 h of treatment with an autophagy activator (Rapamycin, 500 nM). **(F, G)** Grayscale analysis of the LC3B and p62 Western blotting results. Each column being was compared with the control at 0 h. At least three independent experiments were performed. Data are presented as means ± SD, and differences were identified using an unpaired $t$ test (*$P < 0.05$, **$P < 0.01$, ***$P < 0.001$, ns, no significant difference).

IFITM3-RNAi transfection inhibited IFITM3 levels to 50% of that in the infected group, indicating successful *IFITM3* knockdown (Fig 2A and B). Meanwhile, the LC3B II/LC3B I levels decreased and p62 levels increased in *C. parvum*-infected *IFITM3* knockdown cells (Fig 2A, C, and D). The immunofluorescence result showed that LC3B levels increased at 24 h post-infection (hpi), whereas *IFITM3*

knockdown inhibited the expression of LC3B (Fig 2E and F). Beclin-1, autophagy-related (ATG)7, and ATG5 are important autophagy-associated proteins whose expression is elevated during autophagy. Infection of *C. parvum* leads to up-regulation of Beclin-1, ATG7, and ATG5. Knockdown of *IFITM3* in infected cells resulted in further decreases in Beclin-1, ATG7, and ATG5 levels. These results

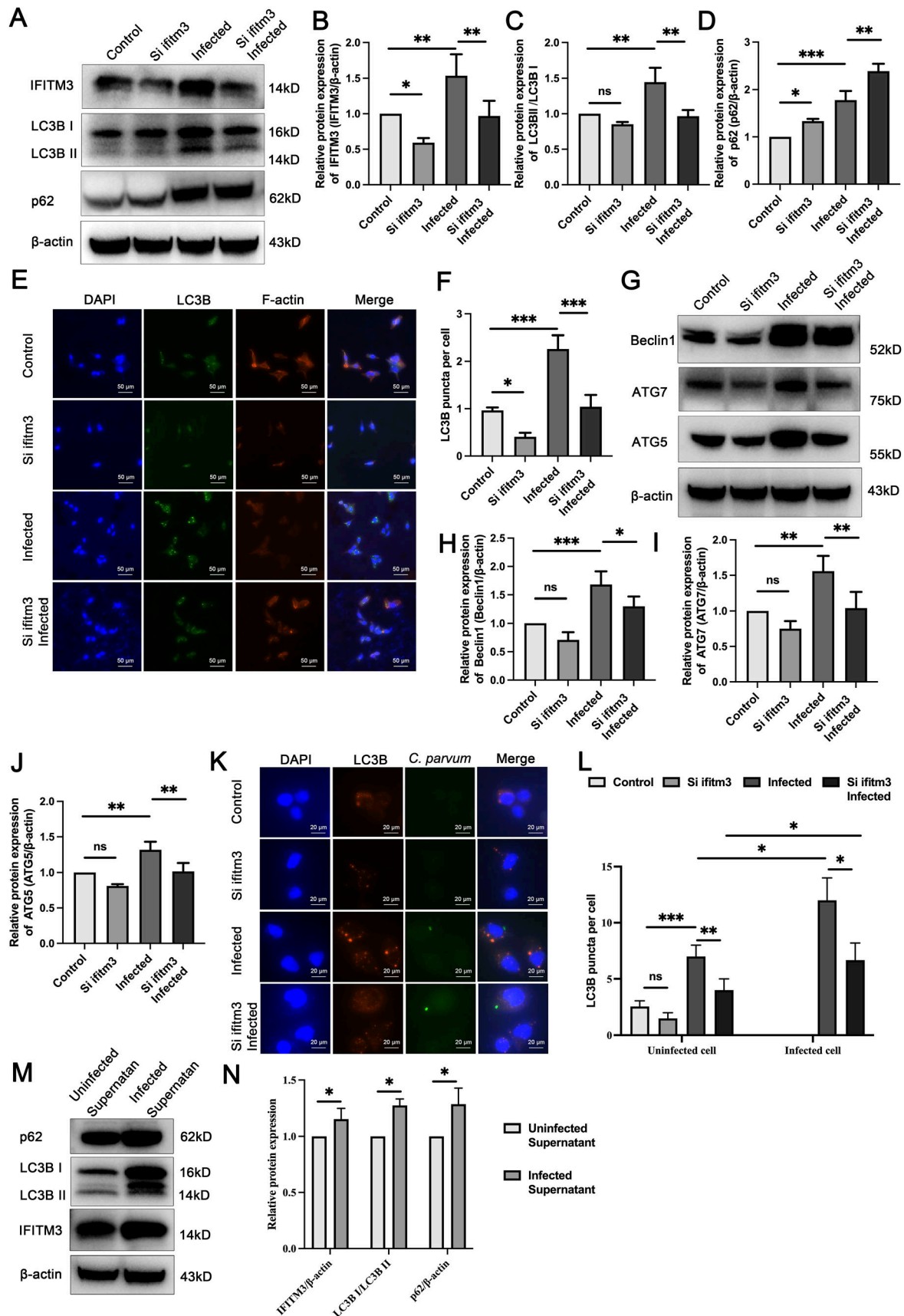

confirmed that IFITM3 positively regulates *C. parvum*-induced autophagy via Beclin-1, ATG7, and ATG5 (Fig 2G–J).

To explore what cells in the images shown Fig 2 are infected and whether the autophagy events occur in these infected cells, *C. parvum* LC3B levels were tested using immunofluorescence. The result showed that autophagy occurred in both infected and uninfected cells with higher levels of autophagy in infected cells (Fig 2K and L). The supernatant of the infected group (24 hpi) and uninfected group was added to HCT-8 cells. It was found that the infected cell culture supernatant could also cause slight increases in IFITM3, LC3, and p62 levels (Fig 2M and N).

### IFITM3 could modulate and combine with LC3B and autophagy-associated proteins to mediate autophagy during *C. parvum* infection

To further characterize the relationship between IFITM3 and autophagy, the inhibitor of autophagy 3-methyladenine (3-MA) and an *IFITM3* overexpression plasmid were used in HCT-8 cells infected with *C. parvum*. The results demonstrated that when autophagy was inhibited, the level of LC3B II/LC3B I decreased, whereas the IFITM3 level showed no obvious difference. Overexpression of *IFITM3* increased the level of IFITM3 substantially and up-regulated the key marker of autophagy, LC3B II/LC3B I (Fig 3A–C). Inhibition of autophagy in infected cells resulted in increased levels of p62 and decreased levels of Beclin-1, ATG7, and ATG5 (Fig 3D–H). Overexpression of *IFITM3* in infected cells could increase the levels of autophagy-associated proteins Beclin-1, ATG7, and ATG5, whereas decreasing the level of p62 (Fig 3D–H). These findings further indicated that IFITM3 regulates autophagy caused by *C. parvum* infection through Beclin-1, ATG7, and ATG5.

Confocal immunofluorescence analysis showed that IFITM3 colocalized with LC3B and this colocalization increased after *C. parvum* infection. The use of the autophagy inhibitor 3-MA decreased the level of LC3B in infected cells, whereas overexpression of *IFITM3* increased the level of LC3B and its colocalization with IFITM3, suggesting that IFITM3 could regulate autophagy in *C. parvum*-infected HCT-8 cells by binding to LC3B (Fig 3I and J). Moreover, the coimmunoprecipitation (Co-IP) results showed that IFITM3 interacted with LC3B, Beclin-1, and ATG5, but not with ATG7 or p62 in *C. parvum*-infected HCT-8 cells (Fig 3K–N).

### IFITM3 could protect HCT-8 cells from *C. parvum* infection by improving the survival rate of cells and reducing *C. parvum* load

IFITM3 has also been extensively studied in the regulation of apoptosis and the immune response. An *IFITM3* knockdown cell model was constructed and co-cultured with *C. parvum*. Apoptosis analysis was performed on these cells using flow cytometry. The analysis showed that the number of dead cells has no significant change (Fig S2A and B) whereas both early and late apoptosis in the *C. parvum*-infected group increased compared with those in the uninfected group (Fig 4A–D). Besides, we observed significantly higher apoptosis rates in the *IFITM3* knockdown group after *C. parvum* infection, perhaps as a result of the inhibited autophagy caused by *IFITM3* knockdown (Fig 4A–D).

IL-8 and TNF-$\alpha$ levels are usually increased after *C. parvum* infection. We observed that *C. parvum* evoked an immune response in HCT-8 cells, indicated by the augmented expression of inflammatory cytokines IL-8 and TNF-$\alpha$ at both the mRNA and protein level (Fig 4E–I). Furthermore, knockdown of *IFITM3* led to diminished expression of IL-8 and TNF-$\alpha$ at both the mRNA and protein level, which might be related to the elimination of the parasite (Fig 4E–I). The results of an enzyme-linked immunosorbent assay (ELISA) showed up-regulated supernatant levels of IL-8 and TNF-$\alpha$ in the infected group, but down-regulated levels the in si*IFITM3* group, especially for IL-8 (Fig 4J and K). To determine the role and impact of IFITM3 in *C. parvum* proliferation, fluorescence detection and quantitative real-time reverse transcription PCR (qRT-PCR) were performed in each group. The number of *C. parvum* was markedly increased in the *IFITM3* knockdown group (Fig 4L and M). Infection with *C. parvum* and knockdown of *IFITM3* did not influence the number of HCT-8 cells, but increased the number of *C. parvum* (Fig 4N and O).

### IFITM3-induced autophagy plays an important role in the immunity against *C. parvum* infection

Flow cytometry analysis of apoptosis in the autophagy inhibited and infected group and in the *IFITM3* overexpression and infected group showed that compared with the *C. parvum*-infected group, early apoptosis and late apoptosis in the autophagy inhibited and infected group were increased (Fig 5A–D). However, the live cell percentage was reduced (Fig 5A–D), suggesting that inhibition of autophagy could reduce the survival rate of cells infected with *C. parvum*. Overexpression of *IFITM3* reduced the early apoptosis and late apoptosis level, and improved the survival rate of cells infected with *C. parvum* (Fig 5A–D).

To identify whether the changes of inflammatory cytokines induced by IFITM3 expression are related to autophagy, the levels of IL-8 and TNF-$\alpha$ were detected by Western blotting and ELISA. Inhibition of autophagy by 3-MA led to diminished levels of IL-8, similar to the result in the *IFITM3* knockdown group. However, the level of TNF-$\alpha$ showed no significant changes (Fig 5E–I). Overexpression of IFITM3 increased the levels of IL-8 and TNF-$\alpha$ (Fig 5E–I). Next, qRT-PCR was performed in each group to investigate the

**Figure 2. Knockdown of *IFITM3* weakens autophagy of HCT-8 cells after *C. parvum* infection.**
**(A)** Representative Western blotting analysis of IFITM3, LC3B, and p62 levels in the control, siIFITM3, infected, and siIFITM3 infected groups. **(B, C, D)** Statistics of the IFITM3 (B), LC3B (C), and p62 (D) Western blotting results. **(E, F)** Fluorescence microscopy imaging and statistics of the number of LC3B puncta in the four groups. **(G)** Representative Western blotting and histogram analysis of Beclin1, ATG7, and ATG5 levels in HCT-8 cells in the four groups. **(H, I, J)** Statistics of the Beclin-1 (H), ATG7 (I), and ATG5 (J) Western blotting results. **(K)** Fluorescence microscopy imaging of the LC3B puncta and *C. parvum* in the four groups. **(L)** Statistics of the number of LC3B puncta in uninfected and infected cells in the four groups. **(M, N)** Western blotting analysis of IFITM3, LC3B, and p62 in the cells cultured with supernatant of infected group (24 hpi) and uninfected group. (Control: uninfected group; siIFITM3: IFITM3-RNAi transfected; Infected: *C. parvum* infected for 24 h; siIFITM3 infected: IFITM3-RNAi transfected and *C. parvum* infected for 24 h). At least three independent experiments were performed. Data are presented as means ± SD, and differences were identified using an unpaired *t* test and ANOVA (*$P < 0.05$, **$P < 0.01$, ***$P < 0.001$, ns, no significant difference).

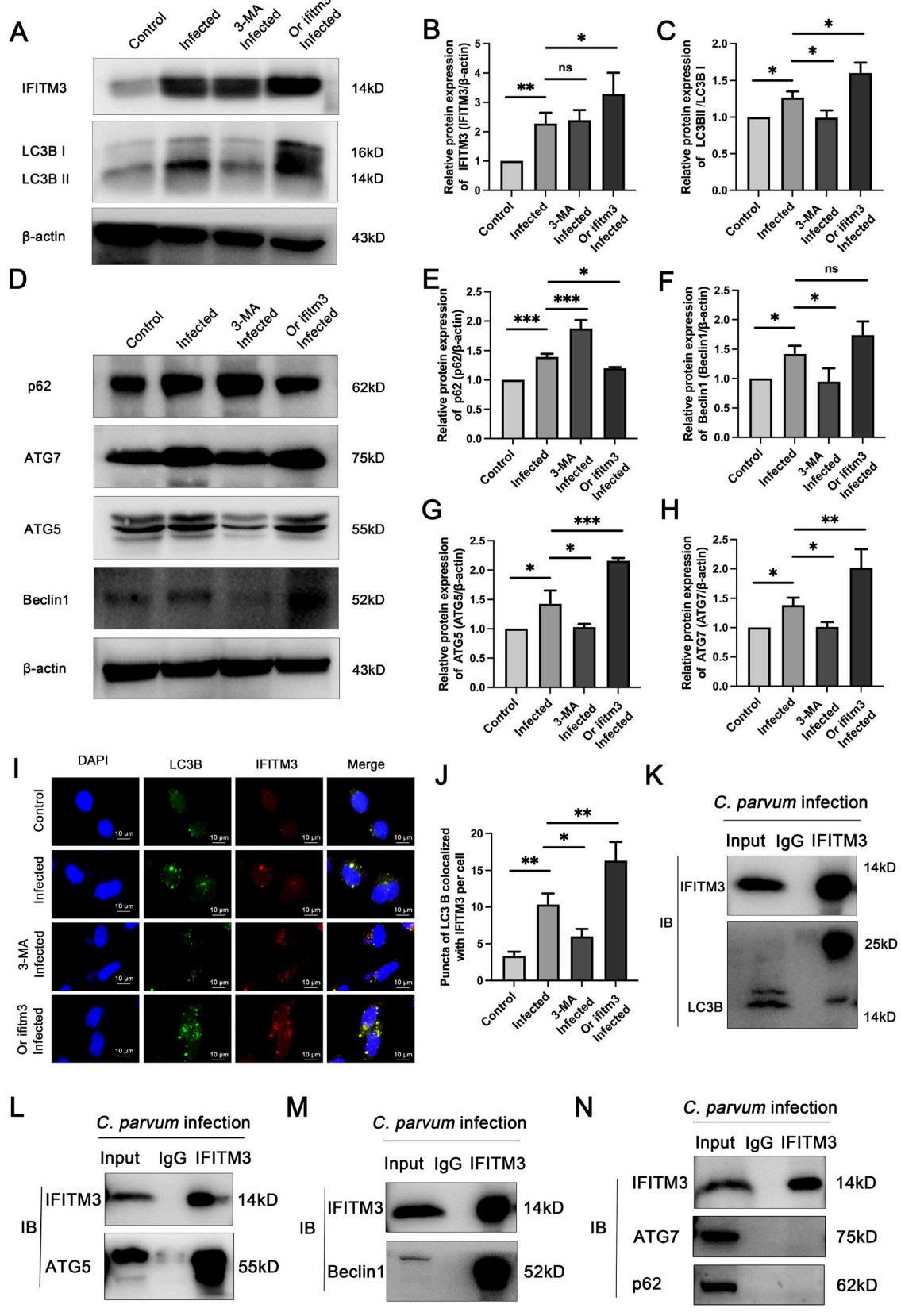

role of autophagy in *C. parvum* proliferation. The number of *C. parvum* was markedly increased in cells treated with 3-MA, whereas it was decreased in *IFITM3* overexpressing cells (Fig 5J and K).

### IFITM3-induced autophagy is regulated by STAT1, a key cell signaling protein

Our previous study found that the level of signal transducer and activator of transcription 1 (STAT1) increased after *C. parvum* infection (Li et al, 2021). To explore the relationship between the STAT1 and IFITM3 expression levels in HCT-8 cells after *C. parvum* infection, the inhibitor of STAT1 Fludarabine (10 µM) was used in uninfected and *C. parvum*-infected HCT-8 cells. Western blotting and qRT-PCR showed that STAT1 expression in the STAT1-inhibitor group was decreased compared with that in uninhibited cells, exhibiting the highest inhibition efficiency (Fig 6A–C and F). We observed that the expression levels of STAT1 and IFITM3 increased during *C. parvum* infection (Fig 6A and D). After inhibiting STAT1, the expression of IFITM3 was decreased in both infected and uninfected cells, and the autophagy level was decreased in infected cells (Fig 6A, E, and G). Infection by *C. parvum* and inhibition of STAT1 did not influence the number of HCT-8 cells, but did increase the number of *C. parvum* (Fig 6H and I).

To verify whether IFITM3 affects STAT1 expression, we used the siRNA to knock down *IFITM3*, which resulted increased levels of STAT1 and p-STAT1 in both infected and uninfected cells (Fig 6J–M). Then, to establish how STAT1 regulates the expression of IFITM3, we performed coimmunoprecipitation to determine whether STAT1 directly interacts with IFITM3. The result showed that there is no direct interaction between STAT1 or p-STAT1 and IFITM3 in *C. parvum*-infected HCT-8 cells (Fig 6N).

## Discussion

Our understanding of the molecular mechanisms of *C. parvum*-host interactions and the underlying factors that govern infectivity and disease pathogenesis is limited. Herein, we revealed that STAT1-IFITM3 regulation of autophagy in epithelial immunity confers resistance to *C. parvum* infection. To study the role of IFITM3 expression during *C. parvum* infection, we examined the expression of IFITM3 in HCT-8 cells, showing that IFITM3 expression of HCT-8 was higher in infected than in uninfected cells.

Autophagy plays an important role in the host response to infection by most intracellular pathogens, including bacteria, viruses, and parasites (Latr & Harnett, 2017; Mao et al, 2019; Xiao & Cai, 2020). Autophagy of HCT-8 cells increases after *C. parvum* infection

(Jiang et al, 2022). Complete cell autophagy was activated during *C. parvum* infection, which reduced the burden of *C. parvum* in HCT-8 cells. *C. parvum* not only induced the formation of a large number of autophagosomes, but also promoted the fusion of these autophagosomes with lysosomes (Wu, 2024). Herein, we showed that *C. parvum*-induced autophagy in HCT-8 cells, but autophagy flow was blocked, paused, or was not induced at 12 h and 24 h. Autophagy of host cells in response to infection by protozoan parasites could clear pathogens or promote parasite proliferation. For example, plasmodium protein UIS3 protects the parasite from autophagy clearance (Yao & Klionsky, 2018). Moreover, autophagic targeting results in *Toxoplasma gondii* killing via the fusion of autophagosomes and lysosomes (Subauste, 2009). IECs defend against *C. parvum* infection by regulating their autophagy and apoptosis through the miR-199a-3p-MTOR axis (Wu, 2024). In the present study, we found that knockdown of *IFITM3* and inhibition of autophagy using 3-MA increased the intracellular proliferation of *C. parvum*, whereas overexpression of *IFITM3* decreased the intracellular proliferation of *C. parvum*. These results suggested that increasing IFITM3 expression and autophagy could reduce the burden of *C. parvum* in HCT-8 cells.

Although the role of IFITM3 in certain viral infections is known, the mechanism of autophagy regulation by IFITM3 in HCT-8 cells infected with *C. parvum* is unclear. Overexpression of *IFITM3* and other ISGs, such as IFN-induced guanylate-binding protein 1 (GBP1) and stimulator of IFN genes (STING), has been reported to induce autophagy during virus infection (Gu et al, 2021; Shi et al, 2018). For example, IFITM3 inhibits Sendai virus-triggered induction of type I IFN by mediating autophagosome-dependent degradation of IRF3. Overexpression of *IFITM3* enhanced the process of autophagy, as evidenced by the transformation of LC3 from type I to type II (Jiang et al, 2018). Seneca virus A (SVA) infection induces autophagy in NCI-H1299 cells to promote replication. Overexpression of IFITM3 triggers the induction of autophagy, again evidenced by increased LC3 II expression and colocalization of LC3 with LAMP1, which correlated positively with enhanced SVA replication (Aftab et al, 2024). In addition to LC3, some autophagy-related proteins can also be used as indicators to detect autophagy, such as: p62, Beclin-1, ATG7, and ATG5 (Deretic, 2021). p62/SQSTM1 binds directly to LC3 via a specific sequence motif and is itself degraded by autophagy. Thus, p62 might be used as a marker to study autophagic flux (Bjørkøy et al, 2009). Beclin-1 is an essential protein for autophagy, including autophagosome biogenesis and maturation (Kimura et al, 2017; Li et al, 2018). Autophagy-related proteins ATG5 or ATG7 promote autophagy and inhibit endoplasmic reticulum (ER) stress (Komatsu et al, 2005; Zheng et al, 2019). In this study, infection by *C. parvum* up-regulated the ratio of LC3B II to LC3B I, p62, Beclin-1, ATG7, and

**Figure 3. IFITM3 combines with autophagy-associated proteins to participate in autophagy caused by *C. parvum* infection.**
**(A)** The inhibitor of autophagy, 3-MA, and the *IFITM3* overexpression plasmid were used in HCT-8 cells infected with *C. parvum*. Representative Western blotting analysis of IFITM3 and LC3B expression levels in uninfected, infected, 3-MA plus infected, and *IFITM3* overexpression infected groups. **(B, C)** Statistics of the LC3B and p62 Western blotting results in the four groups. **(D, E, F, G, H)** Representative Western blotting and histogram analysis of p62 (E), Beclin-1 (F), ATG5 (G), and ATG7 (H) levels in HCT-8 cells in the four groups. **(I, J)** The expression levels and colocalization of IFITM3 and LC3B in the four groups, as assessed using laser confocal microscopy. **(K)** HCT-8 cells were infected with *C. parvum* and then subjected to Co-IP analysis for the association of IFITM3 with LC3B. **(L, M, N)** Co-IP analysis for the association of IFITM3 with ATG5 (L), Beclin-1 (M), ATG7 (N), and p62 (N) in 24-h HCT-8 cells. (Control: uninfected group; Infected: *C. parvum* infected for 24 h; 3-MA infected: autophagy inhibited and *C. parvum* infected for 24 h; or *IFITM3* infected: cells overexpressing *IFITM3* and infected for 24 h with *C. parvum*). At least three independent experiments were performed. Data are presented as means ± SD, and differences were identified using ANOVA (*$P < 0.05$, **$P < 0.01$, ***$P < 0.001$, ns, no significant difference).

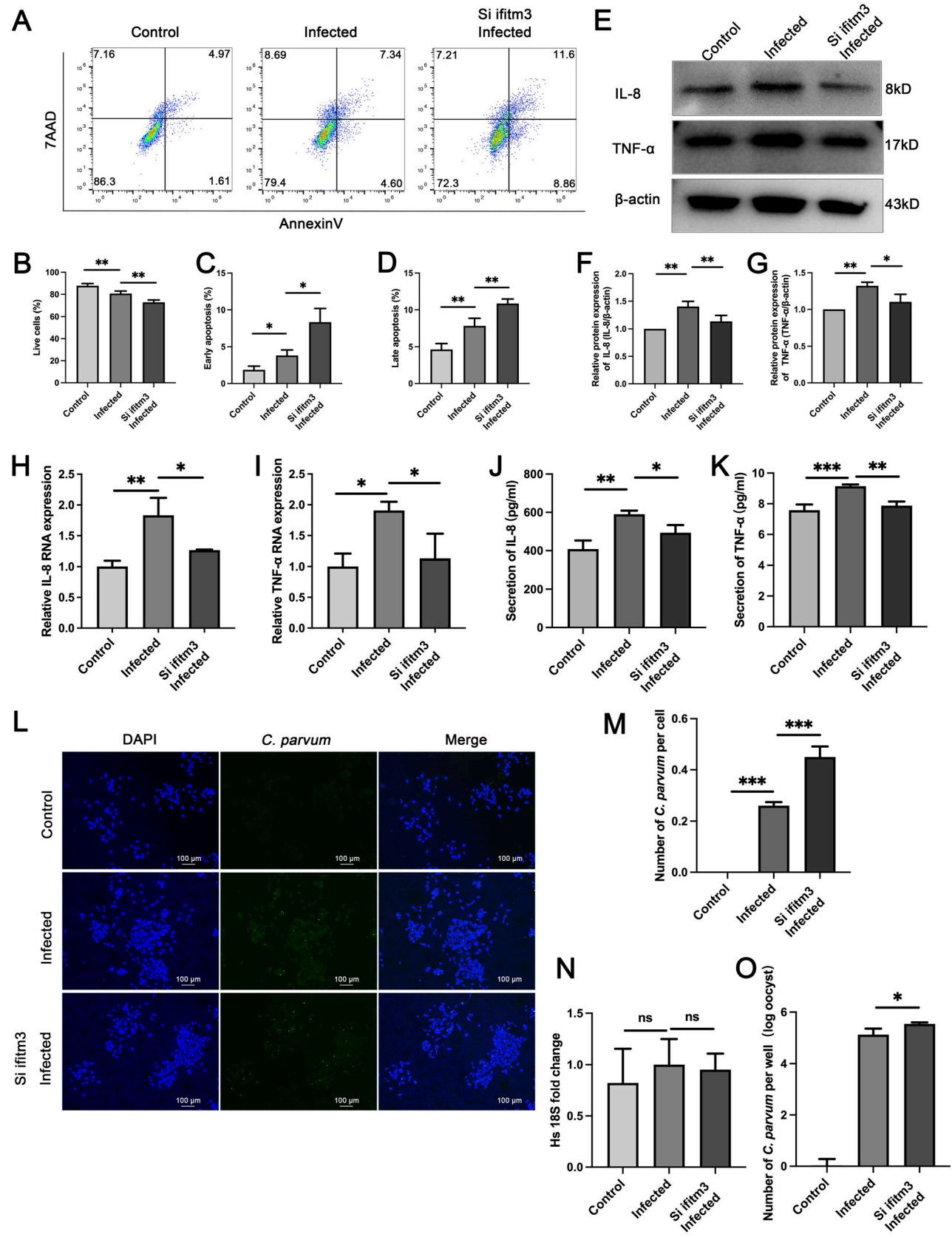

ATG5. Knockdown of *IFITM3* resulted in increased p62 levels but a decreased LC3B II to LC3B I ratio, and Beclin-1, ATG7, and ATG5 levels in infected and uninfected cells. This suggested that IFITM3 could positively regulate *C. parvum*-induced autophagy via Beclin-1, ATG7 and ATG5.

Furthermore, characterization of the relationship between IFITM3 and autophagy showed no obvious difference in the IFITM3 level when autophagy was inhibited. Overexpression of IFITM3 could further improve autophagy, suggesting that IFITM3 can induce autophagy and IFITM3 might act upstream of the autophagosome. Inhibition of autophagy after infection resulted in increased p62 levels and decreased Beclin-1, ATG7, and ATG5 levels, similar to the response to *IFITM3* knockdown, whereas overexpression of *IFITM3* had the opposite effect. These results indicated that IFITM3 controls autophagy through Beclin-1, ATG7, and ATG5. Interestingly, the role of IFITM3 in restricting viruses is closely related to its cellular localization, and it could combine with LC3B and Beclin-1 (Jiang et al, 2018). A study showed that the distribution and clustering of GFP-LC3B was significantly altered by HA-IFITM3 (Yount et al, 2012). Our results demonstrated that IFITM3 colocalized with LC3B after *C. parvum* infection. The autophagy inhibitor decreased this colocalization whereas overexpression of IFITM3 increased it, suggesting that IFITM3 could promote autophagy in *C. parvum*-infected HCT-8 cells by binding to LC3B. Moreover, IFITM3 interacted with LC3B, Beclin-1, and ATG5, but not with ATG7 and p62 in *C. parvum*-infected HCT-8 cells. These observations suggested that IFITM3 might induce autophagy mainly by combining with Beclin-1, ATG5, and LC3B to form a complex and participate in the formation of autophagosomes during *C. parvum* infection.

IFITM3 has also been extensively studied in the regulation of apoptosis and the immune response. MicroRNA miR-29a suppresses the growth and metastasis of hepatocellular carcinoma (HCC) through IFITM3, and knockdown of *IFITM3* promoted apoptosis of HCC cells (Liang et al, 2018). IFITM3 promotes bone metastasis of prostate cancer cells by mediating activation of the TGF-β signaling pathway; however, knockdown of *IFITM3*-induced apoptosis and inhibited their migration (Liu et al, 2019). Herein, we found that the apoptosis rate increased in *C. parvum*-infected HCT-8 cells. Moreover, we observed significantly higher rates of apoptosis in the *IFITM3* knockdown group after *C. parvum* infection. The functional relationship between apoptosis and autophagy is complex. Under certain circumstances, autophagy constitutes a stress adaptation that suppresses apoptosis (Maiuri et al, 2007; Prerna, 2022). Our study showed that compared with the *C. parvum*-infected group, the apoptosis level was increased after inhibition of autophagy whereas overexpression of *IFITM3* could reduce the apoptosis level of cells infected with *C. parvum*. Taken together,

these results suggested that IFITM3 might inhibit apoptosis and enhance cell survival after *C. parvum* infection by inducing autophagy.

Furthermore, *C. parvum* infection-mediated stimulation of TNF-α and IL-8 secretion from IECs could significantly reduce *C. parvum* development (Seydel et al, 1998; Zhang, 2024). In this study, *C. parvum*-induced an immune response in HCT-8 cells, indicated by augmented expression of inflammatory cytokines IL-8 and TNF-α. Knockdown of *IFITM3* reduced the expression levels of IL-8 and TNF-α. Autophagy mediates hepatitis B virus protein HBx-induced NF-κB activation and the release of IL-6, IL-8, and C-X-C motif chemokine ligand 2 (CXCL2) in hepatocytes (Luo et al, 2015). Moreover, autophagy is required for TLR-mediated IL-8 production in IECs (Li et al, 2011). *T. gondii* infection in ATG5[ΔIEC] mice demonstrated that Paneth cell-intrinsic autophagy prevents TNF-α and IFN-γ-driven intestinal cell death (Burger et al, 2018). In the present study, inhibition of autophagy led to reduced expression of IL-8, which was similar to the result in *IFITM3* knockdown cells. Overexpression of *IFITM3* could increase the levels of IL-8 and TNF-α. This suggested that IFITM3 regulation of autophagy in *C. parvum*-infected HCT-8 cells promotes the secretion of inflammatory factor IL-8, which plays key roles in resisting *C. parvum* infection. However, TNF-α appears to have no connection with autophagy. Taken together, our results show that complete cell autophagy was activated by IFITM3 during *C. parvum* infection, which inhibited apoptosis, increased the inflammatory response, and reduced the burden of *C. parvum* in HCT-8 cells.

IFITM3 expression is regulated by many transcription factors. In a tumor model, depletion of IFITM3 in regulatory T cells (Tregs) enhanced the translation and phosphorylation of STAT1. By contrast, decreased IFITM3 expression in STAT1-deficient Tregs indicated that STAT1 regulates the expression of IFITM3 to form a feedback loop (Liu et al, 2024). In colorectal cancer, phosphorylated CTD interacting factor 1 (PCIF1) modulates colorectal cancer growth and response to anti-programmed cell death 1 (PD-1) in a context-dependent mechanism in which PCIF1 directly targets FOS, IFITM3, and STAT1 via N6,2'-O-dimethyladenosine (m6Am) modifications. During immunotherapy, the PCIF2-FOS-TGF-β and PCIF1-STAT/IFITM3-IFN-γ axes contribute to resistance to anti-PD-1 therapy (Wang et al, 2023). The chicken-derived MER receptor tyrosine kinase (MERTK) protein (chMertk) can phosphorylate STAT1. Then, p-STAT1 combines with STAT2 and IRF9 to form the ISGF3 complex, which then translocates into the nucleus where it binds to IFITM3 (Tan et al, 2023). Furthermore, *C. parvum* parasites exclusively infect epithelial cells and the ability of IFNs to activate the transcription factor STAT1 in IECs is required for parasite clearance (Pardy et al, 2024). Herein, we observed that the levels of STAT1 and p-STAT1

**Figure 4. IFITM3 could improve the survival rate of cells after infection and protect HCT-8 cells from *C. parvum* infection.**
**(A)** Apoptosis was assessed using Annexin V and 7-Amino-Actinomycin staining, and apoptotic cells were identified by flow cytometry in uninfected, infected, and siIFITM3 infected groups. **(B, C, D)** Histogram analysis of cell viability of live cells (B), early apoptosis (C), and late apoptosis (D) cells in the three groups. **(E, F, G)** Western blotting results and histogram analysis of IL-8 (F) and TNF-α (G) protein levels in the three groups. **(H, I)** qRT-PCR analysis of *IL-8* and *TNFA* expression levels in the three groups of HCT-8 cells. **(J, K)** ELISA analysis of IL-8 and TNF-α levels in the supernatants of the three groups. **(L, M)** Representative fluorescence microscopy result of *C. parvum* numbers and histogram analysis of *C. parvum* numbers per cell in the three groups. **(N, O)** Parasite numbers and cell numbers were quantified by qRT-PCR assays at 24 hpi. (Control: uninfected group; Infected: *C. parvum*-infected group; siIFITM3 infected: IFITM3-RNAi transfected and *C. parvum*-infected group). At least three independent experiments were performed. Data are presented as means ± SD, and differences were identified using ANOVA (*$P < 0.05$, **$P < 0.01$, ***$P < 0.001$, ns, no significant difference).

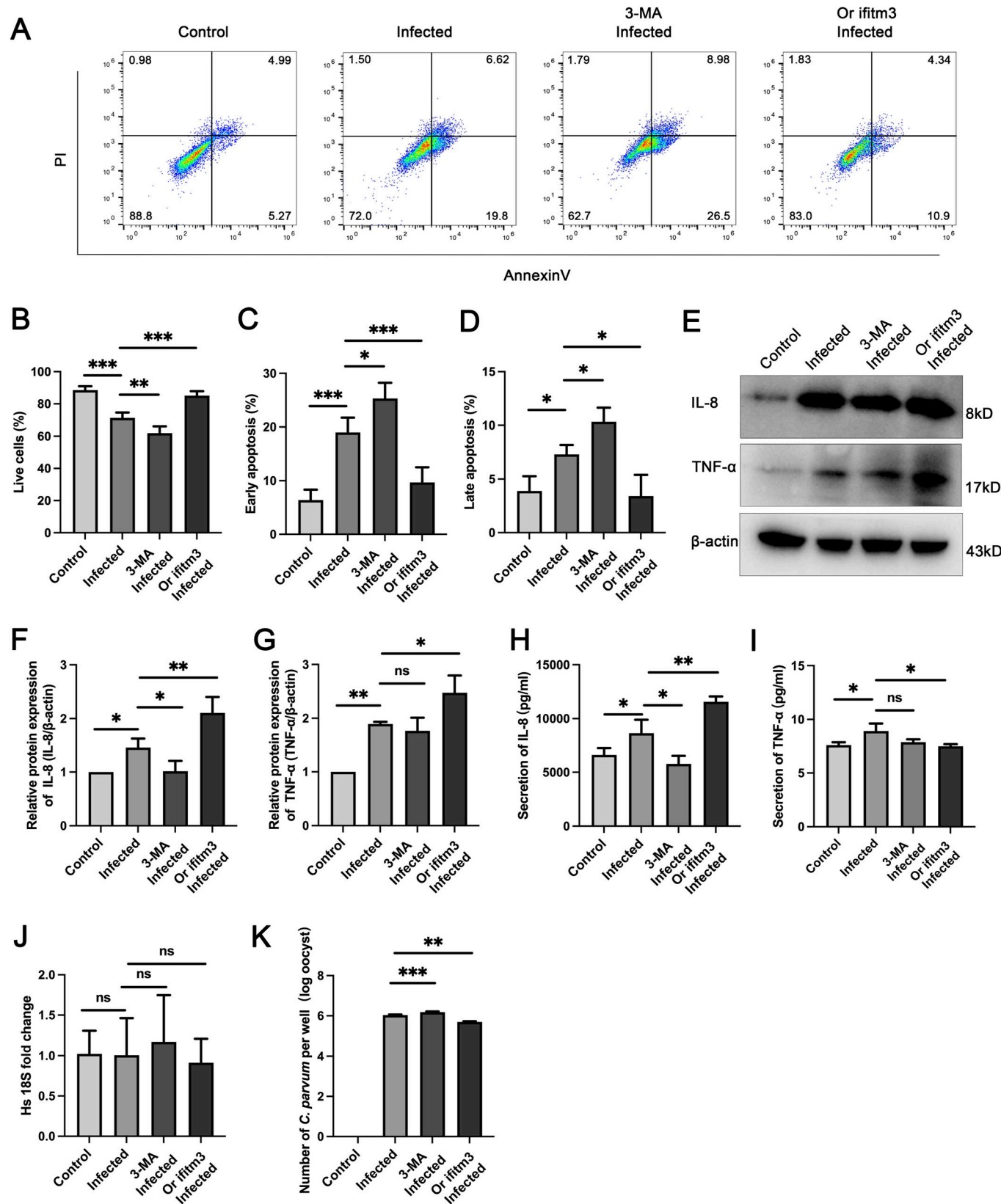

increased during *C. parvum* infection. Inhibition of STAT1 down-regulated IFITM3, inhibited autophagy, and further increased the parasite burden. Knockdown of *IFITM3* resulted in increased levels of STAT1 and p-STAT1. However, we detected no interaction between STAT1 or p-STAT1 and IFITM3 in infected HCT-8 cells. Despite this finding, we believe that STAT1 is an essential factor for IFITM3 expression, and IFITM3 and STAT1 might form a feedback loop to maintain the activation of autophagy.

In conclusion, the results presented here identified increased activation of the STAT1-IFITM3 axis caused by *C. parvum* infection, thus promoting autophagy, inhibiting apoptosis, and increasing the inflammatory response to reduce the parasite load. These responses play a positive role in controlling *C. parvum* infection and promoting host cell survival. Our findings provide a new perspective for the prevention and control of *C. parvum*. Nevertheless, the detailed mechanism of IFITM3 regulation of autophagy induced by *C. parvum* infection and how autophagy affects the outcome of *C. parvum* infection still require further research.

# Materials and Methods

## Preparation of *C. parvum* oocysts

*C. parvum* oocysts were purchased from Waterborne Inc. An appropriate amount of *C. parvum* oocysts was immersed in ice-cold PBS solution containing 20% sodium hypochlorite, pretreated for 10 min at 4°C, and centrifuged at 3,667$g$ for 10 min. The supernatant was discarded and the oocysts were re-suspended in PBS. After two washes with PBS, the oocysts were incubated for 30 min at 37°C with 0.75% sodium taurocholate (Sigma-Aldrich) to induce excystation, and then re-suspended in complete medium for the cell infection test.

## Culture and infection of HCT-8 cells

The HCT-8 cell line (ATCC) was stored in our laboratory. Frozen cell stocks were defrosted in a 37°C water bath with shaking. HCT-8 cells were cultured in DMEM nutrient mix F-12 containing 10% FBS, 100 IU/ml streptomycin, and 100 IU/ml penicillin (All from Gibco). Cells for the STAT1-inhibitor plus *C. parvum* infection group were incubated with 10 $\mu$M Fuldarabine (MCE). Furthermore, 1 mM of autophagy inhibitor 3-MA (MCE) was tested in cells of the autophagy-inhibitor plus *C. parvum* infection group. For infection, HCT-8 cells were co-cultured with oocysts in a 2:1 ratio (oocysts: cells) and incubated in a 37°C incubator containing 5% $CO_2$. At 24 h after infection, cells and oocysts were observed under a Nikon Eclipse Ti inverted microscope (Nikon), and the HCT-8 cells were collected.

**Table 1. siRNA duplexes used to knock down human IFITM3.**

| Gene | siRNA sequence 5′ to 3′ |
|---|---|
| Nontarget siRNA | Sense: 5′-UUCUCCGAACGUGUCACGUTT-3′ |
| | Antisense: 5′-ACGUGACACGUUCGGAGAATT-3′ |
| *IFITM3* siRNA | Sense: 5′-TGCTGATCTTCCAGGCCTA -3′ |
| | Antisense: 5′-TCGTCTGGTCCCTGTTCAA-3′ |

## Plasmid transfection

The *IFITM3* overexpression plasmid was constructed by Shanghai Genechem Co., Ltd. Transfection was initiated when the HCT-8 cells grew to more than 60% confluence. The transfection system comprised: dilution of 4 $\mu$l of Lipofectamine 2000 (Invitrogen) with 100 $\mu$l of Opti-MEM (Thermo Fisher Scientific) per sample well, inversion, gentle mixing, and incubation at RT for 5 min. Then, 1.6 ng of IFITM3 plasmid was diluted with 100 $\mu$l of Opti-MEM per sample well and cultured at RT for 5 min to form the transfection complex. The original medium of the cells was discarded, 200 $\mu$l transfection complex was added to each well, and 800 $\mu$l of Opti-MEM was added for complete culture. The culture medium was changed 6 h later. According to the experimental requirements, qRT-PCR was performed 24 h after transfection to detect mRNA, and protein levels were detected 48 h later using Western blotting.

## RNA interference (RNAi) constructs

An siRNA targeting *IFITM3* and nontarget siRNA were produced RiboBio Co., Ltd. (Guangzhou, China), and the sequences are shown in Table 1. The siRNAs were dissolved in RNase-free water to prepare a 20-$\mu$M stock solution. Cells were seeded in a 24-well plate and grown to 30–50% confluence (~24 h). Then, 1.25 $\mu$l of siRNAs in storage solution were separately diluted in 30 $\mu$l of ribo FECT CP buffer (RiboBio) and mixed. Then, 3 $\mu$l of ribo FECT CP reagent (RiboBio) were added to the mixture, and the transfection complex was prepared by incubation at RT for 10 min. The transfection complex was separately diluted in 465.75 $\mu$l DMEM F-12 containing 10% FBS, but without penicillin/streptomycin, and incubated at RT for 20 min. Finally, the diluted transfection complex was added to the cells in the 24-well plate. The cells were cultured for 48 h for transfection.

## Western blotting

The cells from each group were lysed using ice-cold radio-immunoprecipitation lysis buffer (RIPA, Beyotime) and phenyl-methanesulfonyl fluoride (PMSF, Beyotime). Then, the cells were incubated on ice for 30 min, followed by centrifugation for 20 min at

---

**Figure 5. IFITM3-induced autophagy plays an important role in immunity against *C. parvum* infection.**
**(A)** Apoptosis was assessed by Annexin V and PI staining, and apoptotic cells were identified by flow cytometry in uninfected, infected, 3-MA infected, and *IFITM3* overexpression infected groups. **(B, C, D)** Histogram analysis of cell viability of live cells (B), early apoptosis (C), and late apoptosis (D) cells in the four groups. **(E, F, G)** Western blotting results and histogram analysis of the protein levels of IL-8 (F) and TNF-$\alpha$ (G) in the four groups. **(H, I)** ELISA analysis of IL-8 and TNF-$\alpha$ levels in the supernatants of the four groups. **(J, K)** Parasite numbers and cell numbers were quantified by qRT-PCR assays at 24 hpi. (Control: uninfected group; infected: *C. parvum*-infected group; 3-MA infected: autophagy inhibited and *C. parvum*-infected group; or *IFITM3* infected: *IFITM3* overexpression and *C. parvum*-infected group). At least three independent experiments were carried out. Data are presented as means ± SD, and differences were identified using ANOVA (*$P < 0.05$, **$P < 0.01$, ***$P < 0.001$, ns, no significant difference).

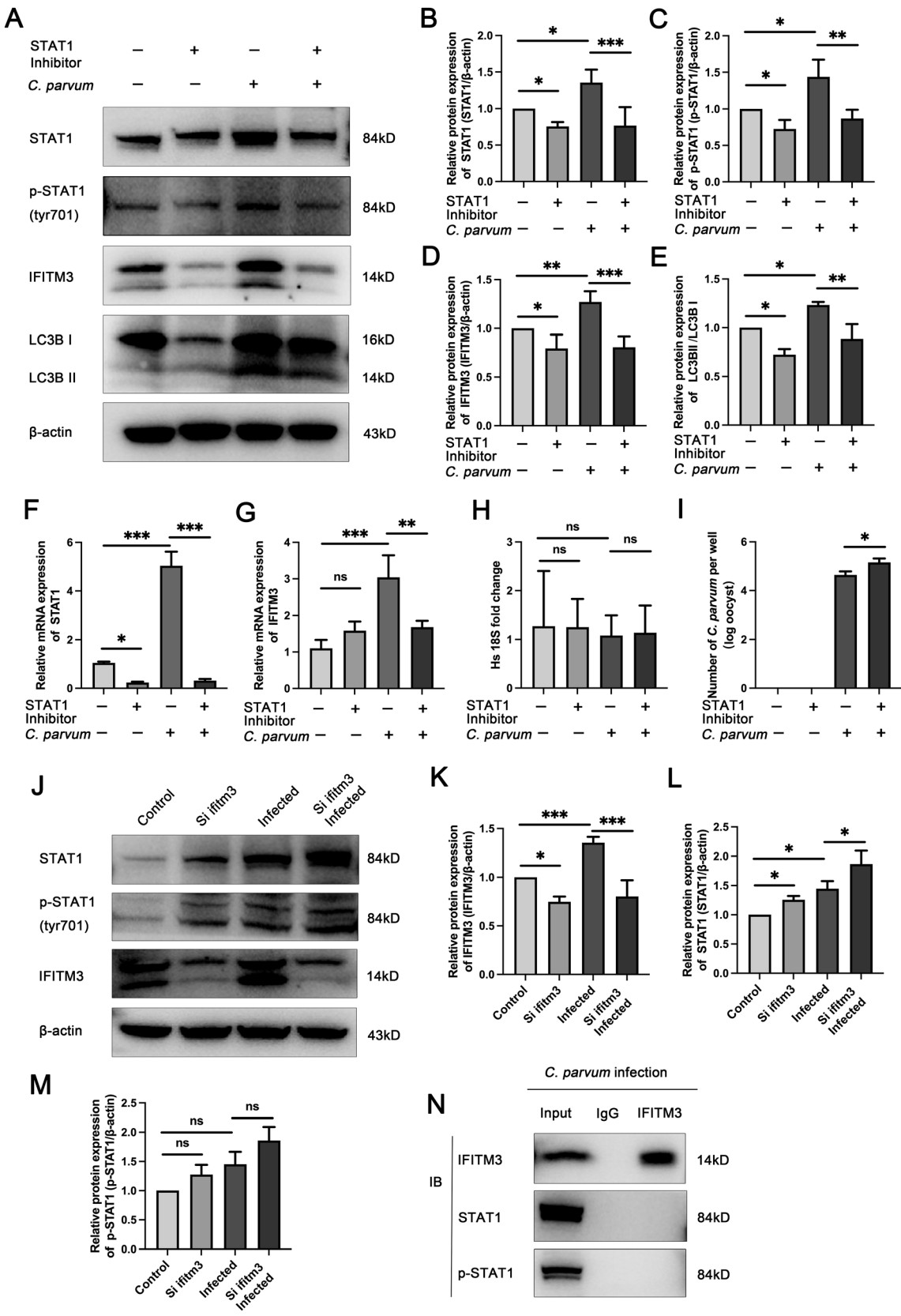

12, 000$g$ and 4°C to remove the debris. The protein concentration of the extract was determined using a bicinchoninic acid (BCA) protein assay kit (Solarbio). Equal amounts of protein were boiled in 5× loading buffer (Beyotime) followed by heating at 95°C for 5 min. The total protein (30 $\mu$g) was separated using 12% sodium dodecyl sulfate-polyacrylamide gel electrophoresis (SDS–PAGE, Beyotime) and transferred to polyvinylidene difluoride (PVDF) membranes (Merck Millipore). The membranes were blocked for 1 h in protein-free rapid blocking buffer (EpiZyme) at RT. Then, the PVDF membranes were incubated overnight at 4°C with primary antibodies recognizing IFITM3 (Proteintech), LC3B (Abcam), p62 (Abcam), ATG5 (Abcam), ATG7 (Abcam), $\beta$-actin (ACTB) (Proteintech), STAT1 (Proteintech), Phospho(p)-STAT1 (Tyr701) (Proteintech), IL-8 (Abcam), and TNF-$\alpha$ (Proteintech). The membranes were then incubated with appropriately labeled secondary antibodies for 2 h at RT. The immunoreactive protein bands were then detected using an enhanced chemiluminescence kit (ECL, EpiZyme). Grayscale analysis to determine the protein expression levels was carried out using ImageJ (NIH), and the relative protein expression level was obtained by normalization to that in the control group. At least three independent samples in each group were counted.

## Immunofluorescence

HCT-8 cells from each group were air-dried onto polylysine microscope adhesion slides in a 12-well cell culture plate for further treatment. After fixation in 4% paraformaldehyde for 30 min, cells were washed three times with PBS, and then permeabilized using 1% Triton-100. The slides were blocked using goat serum for 1 h at RT, followed by incubation overnight at 4°C with anti-IFITM3 and anti-LC3B antibodies, respectively. After washing three times with PBS, the cells were incubated with appropriate concentrations of secondary antibodies for 1 h at 37°C. For cytoskeleton staining, 200 $\mu$l of iFluor 546-Ghost cyclic peptide F-actin (Creative Biolabs) working solution was added to the cells and stained at RT for 15 min, followed by PBS washing. Subsequently, a drop of antifading mounting medium with the nucleic acid stain DAPI (Solarbio) was added to seal the slide. Finally, cell immunofluorescence was assessed using the Invitrogen EVOS FL Auto Cell Imaging System or a Nikon eclipse Ti Confocal Microscope.

## qRT-PCR

Total RNAs from each group were extracted using the TRIZOL reagent according to the manufacturer's protocol (Takara). The primers comprised custom-synthesized products from EnzyArtisan Biotech Co., Ltd., and their sequences are shown in Table 2. cDNAs were prepared from total RNA using a PrimeScript RT–PCR

**Table 2.** List of primers used for qRT-PCR analysis.

| Gene | Primer sequence 5′ to 3′ |
| --- | --- |
| ACTB | Forward: CACCATTGGCAATGAGCGGTTC |
| | Reverse: AGGTCTTTGCGGATGTCCACGT |
| TNF-α | Forward: CTCTTCTGCCTGCTGCACTTTG |
| | Reverse: ATGGGCTACAGGCTTGTCACTC |
| IL-8 | Forward: CTTCCTGATTTCTGCAGCTCTG |
| | Reverse: TGGTCCACTCTCAATCACTCAG |
| IFITM3 | Forward: GTGCTGATCTTCCAGGCCTATG |
| | Reverse: TGGAGTACGTGGGATACAGGTCATR |
| C. parvum-18s | Forward: CGAAGACGATCAGATACCGTCG |
| | Reverse: ACAACCTCCAATCTCTAGGTGGC |
| Hs-18s | Forward: ACT CAA CAC GGG AAA CCT CAC |
| | Reverse: AGC TTA TGA CCC GCA CTT ACT GG |
| STAT1 | Forward: ATGGCAGTCTGGCGGCTGAATT |
| | Reverse: CCAAACCAGGCTGGCACAATTG |

Kit (EnzyArtisan). Then, the cDNA amplification reactions were carried out using a SYBR qPCR Mix (EnzyArtisan). The qRT-PCR analysis reactions were performed on a LightCycler 96 instrument (Roche). The comparative threshold cycle ($2^{-\Delta\Delta Ct}$) method (Livak & Schmittgen, 2001) was used to evaluate the relative mRNA expression, and the *ACTB* gene was used as a normalization control.

## Apoptosis detection

Pretreated cells were digested using 0.25% trypsin to obtain a cell suspension, which was re-suspended in binding buffer after washing with pre-cooled PBS, followed by adjusting the cell concentration to $1 \times 10^6$/ml. Then, 100 $\mu$l of the cell suspension was added with 5 $\mu$l of phycoerythrin (PE) Annexin V and 7-aminoactinomycin D (7-AAD) or propidium iodide (PI), gently shaken, and incubated at RT for 15 min away from light. Each sample was added with 400 $\mu$l of binding buffer, mixed, and then flow cytometry detection was performed using a BD FACSLyric Flow Cytometer (BD Biosciences). Annexin V$^+$/7-AAD$^-$ or Annexin V$^+$/PI$^-$ indicates early apoptosis. Annexin V$^+$/7-AAD$^+$ or Annexin V$^+$/PI$^+$ indicates late apoptosis.

## Calculation of cell numbers and *C. parvum* growth

The proliferation of *C. parvum* oocysts was quantified using SYBR-green qPCR detection using Cp-18S and Hs-18S primers. Standard

**Figure 6. IFITM3-induced autophagy is regulated by STAT1, a key cell signaling protein.**
**(A)** An inhibitor of STAT1, Fludarabine (10 $\mu$M), was used in uninfected and *C. parvum*-infected HCT-8 cells. Western blotting analysis of STAT1, p-STAT1, IFITM3, and LC3B levels in uninfected, STAT1 inhibited, infected, and STAT1 inhibited infected groups. **(B, C, D, E)** Histogram analysis of STAT1 (B), p-STAT1 (C), IFITM3 (D), and LC3B (E) in the four groups. **(F, G)** qRT-PCR analysis of *STAT1* and *IFITM3* expression levels in the four groups. **(H, I)** Parasite numbers and cell numbers were quantified by qRT-PCR assays at 24 hpi. **(J)** Western blotting analysis of STAT1, p-STAT1 and IFITM3 levels in the control, siIFITM3, infected, and siIFITM3 infected groups. **(K, L, M)** Histogram analysis of IFITM3 (K), STAT1 (L), and p-STAT1 (M) in the four groups. **(N)** Coimmunoprecipitation to determine whether IFITM3 directly interacted with STAT1. At least three independent experiments were carried out. Data are presented as means ± SD, and differences were identified using ANOVA (*$P < 0.05$, **$P < 0.01$, ***$P < 0.001$, ns, no significant difference).

curves were established by seeding serially diluted *C. parvum* oocysts ($1 \times 10^1$ to $1 \times 10^5$ oocysts/sample) into negative cells, followed by the same DNA isolation and qRT-PCR techniques used for the experimental samples. The means of cycle threshold ($\Delta C_T$) between $C_{T[Cp18S]}$ and $C_{T[Hs18S]}$ values from technical and biological replicates were plotted against the logarithm of the numbers of inoculated oocysts. Linear regression was conducted to obtain the standard curve, slope value, and $R^2$ value to determine the detection efficacy and data coefficient. The absolute oocyst counts were calculated based on $\Delta C_T$ values and standard curves (Fig S3A). The procedure is briefly described in Supplemental Data 1.

### Fluorescent microscopy detection of *C. parvum*

A Sporo-Glo kit (Waterborne) was used in conjunction with the Focus Detection Method (FDM) to evaluate the viability of *C. parvum* oocysts. HCT-8 cells infected with *C. parvum* on cover glasses in a 12-well cell culture plate were fixed in methanol for 8 min. After the methanol was drained out of each well, 250 $\mu$l of DB Buffer included in the kit was added to cell monolayer and incubated for 30 min. The buffer was removed of each well and a working dilution of antibody reagent was applied to completely cover the cell monolayer. The slide was incubated in a humid chamber at RT for at least 45 min and then the antibody reagent was removed from each well. The slides were rinsed by applying PBS drop wise to cover the monolayer for 3 min. The PBS was removed from each well and the chamber walls were removed. The slide was next be mounted with one drop of antifade mounting medium with DAPI and covered with a coverslip. Finally, confocal microscopy of the cells was performed using the Invitrogen EVOS FL Auto Cell Imaging System.

### ELISA

The cellular supernatant of each group was centrifuged at 1,000$g$ for 10 min to remove impurities, such as cells. The IL-8 and TNF-$\alpha$ levels in the cellular supernatant were assayed using ELISA kit (Multi Science), and the absorbance was measured at OD450 nm.

### Coimmunoprecipitation (Co-IP)

Infected cells from six-well cell culture plates were lysed in immunoprecipitation (IP) lysis/rinsing buffer (25 mM Tris, 150 mM NaCl, 1 mM EDTA, 1% NP40, 5% glycerol) with a protease inhibitor cocktail (Beyotime). The protein concentration was measured using the BCA kit to unify the amount of protein. A Pierce coimmunoprecipitation kit (Thermo Fisher Scientific) was used for IFITM3 IP. Cell lysates from each sample were combined with 4 $\mu$g of IFITM3 IP antibody (Proteintech) overnight at 4°C. Then, 25 $\mu$l of A/G magnetic beads were added to the lysates and incubated for 1 h at RT. The beads were collected using a magnetic stand, and then washed with 500 $\mu$l of IP lysis/wash buffer twice. Next, 500 $\mu$l of ultrapure water was added, the beads were collected on the magnetic stand, and the supernatant was discarded. Subsequently, 100 $\mu$l of elution buffer was added to the beads for IFITM3 elution, and 10 $\mu$l of neutralization buffer was added after 10 min of incubation. Finally, 5 × loading buffer was added to the samples, which were then heated for 5 min at 95°C before SDS–PAGE and Western blotting analysis.

### Statistical analysis

Data derived from at least three experiments are reported as the means ± SD. Differences between two or more groups were determined using an unpaired *t* test or one-way analysis of variance (ANOVA). Statistical analyses and graph preparation were conducted using GraphPad Prism 9 Software (GraphPad Inc.), and the error bars were derived from at least three individual experiments respectively. Levels of significance are indicated as *$P < 0.05$, **$P < 0.01$, ***$P < 0.001$.

## Data Availability

The data reported in this study are included in the article. Further inquiries can be directed to the corresponding author.

## Supplementary Information

## Acknowledgements

We are grateful to Yuan Hu and Hao Zhou at the National Institute of Parasitic Diseases, Chinese Center for Disease Control and Prevention (Chinese Center for Tropical Diseases Research) for their assistance with the experiments. This work was supported by the National Natural Science Foundation of China (grant numbers 82272369 to J Cao, 82372283 to Y Shen) and the Three-Year Initiative Plan for Strengthening Public Health System Construction in Shanghai (2023-2025) Key Discipline Project (grant number GWVI-11.1-12 to J Cao). The funders had no role in the study design, data collection and analysis, decision to publish, or preparation of the manuscript.

### Author Contributions

L Cui: conceptualization, data curation, formal analysis, investigation, and writing—original draft, review, and editing.
T Li: conceptualization, data curation, formal analysis, investigation, visualization, and writing—review and editing.
J Zhang: data curation and formal analysis.
Y Shen: formal analysis and methodology.
J Cao: resources, formal analysis, supervision, funding acquisition, investigation, project administration, and writing—review and editing.

### Conflict of Interest Statement

The authors declare that they have no conflict of interest.

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
