## [Reviewer comments · Life Science Alliance]

Life Science Alliance

STAT1-IFITM3 promotes autophagy in epithelial cells to control *Cryptosporidium parvum* infection

Lijun Cui, Teng Li, Jing Zhang, Yujuan Shen, and Jianping Cao

DOI: <https://doi.org/10.26508/lsa.202503200>

Corresponding author(s): *Jianping Cao, National Institute for Parasitic Diseases*

Review Timeline:

Submission Date:	2025-01-02
Editorial Decision:	2025-02-28
Revision Received:	2025-04-29
Editorial Decision:	2025-06-16
Revision Received:	2025-06-24
Accepted:	2025-06-25

Scientific Editor: Tim Fessenden

Transaction Report:

February 28, 2025

Re: Life Science Alliance manuscript #LSA-2025-03200-T

Prof. Jianping Cao
National Institute of Parasitic Diseases, Chinese Center for Disease Control and Prevention, (Chinese Center for Tropical Diseases Research)
Key Laboratory of Parasite and Vector Biology
No. 207, Rui Jin 2nd Road
Shanghai 200025
China

Dear Dr. Cao,

Thank you for submitting your manuscript entitled "STAT1-IFITM3 promotes autophagy in epithelial immunity to control *Cryptosporidium parvum* infection" to Life Science Alliance. The manuscript was assessed by expert reviewers, whose comments are appended to this letter. We invite you to submit a revised manuscript addressing the Reviewer comments.

Thank you for this interesting contribution to Life Science Alliance. We are looking forward to receiving your revised manuscript.

Sincerely,

B. MANUSCRIPT ORGANIZATION AND FORMATTING:

Reviewer #1 (Comments to the Authors (Required)):

Epithelial cell-intrinsic defenses, particularly those activated by interferon signaling, play a crucial role in *Cryptosporidium* clearance, though the underlying regulatory mechanisms are not yet fully understood. IFITM3 is an interferon-stimulated gene that modulates viral infection through autophagy; however, its role in *C. parvum* infection is unknown. This study reports a significant upregulation of IFITM3 expression following *C. parvum* infection in human HCT-8 cells, which may regulate the cell inflammatory response, cell survival, and parasite clearance through the STAT1-IFITM3 axis. In-depth mechanistic details are explored, including the involvement of autophagy. These findings provide new insights into current understanding of how interferon signaling contributes to cell-intrinsic defense to resist *C. parvum* infection, which is relevant to the development of potential therapeutic targets. The experiments were generally well-performed, and the data appear to be very solid. The manuscript is well-organized and should be of interests to the readers of the journal. My concerns/comments are very minor, as listed below.

1. It would be beneficial to include a discussion on how *C. parvum* infection may induce IFITM3 expression. Specifically, does the infection activate IFN signaling in HCT-8 cells?
2. Cell death through apoptosis could serve as a defense mechanism to clear certain intracellular pathogens. It is particularly interesting that IFITM3 induction can inhibit apoptosis while also enhancing parasite clearance in HCT-8 cells. *C. parvum* infection may induce apoptosis in both directly infected cells and bystander non-infected cells. The observed decrease in infection burden may simply resulted from the increased survival of bystander non-infected cells. The methods used to measure and calculate the infection burden should be clarified and discussed.
3. Some paragraphs in the discussion section are quite lengthy and resemble a description of the results rather than a discussion. These should be shorted for clarity and focus.
4. There are several typos in the manuscripts, such as TNF-a (rather than TNF-) in Figures.

Reviewer #2 (Comments to the Authors (Required)):

In their manuscript title "STAT1-IFITM3 promotes autophagy in epithelial immunity to control *Cryptosporidium parvum* infection", Cui et al. build on recent work in the immunity to *Cryptosporidium* field to show that IFITM3, an interferon stimulated gene important for anti-viral defense and induction of autophagy, can promote control of *Cryptosporidium* in the human colonic adenocarcinoma HCT-8 cell line and is induced downstream of STAT1. This work has the potential to increase our understanding of how autophagy is induced in cultures infected with *Cryptosporidium* and suggest a novel role for the interferon stimulated gene IFITM3 in restriction of parasite growth.

In principle, these studies are of interest to the *Cryptosporidium* community, but there are questions about how the data are presented that make it difficult to assess the manuscript. Below is an extensive list of question but it is hoped that most of these can be addressed with the data sets in hand and not necessarily require new experiments. A theme for some of these is the need for more rigor to measure parasite burden relative to an input value or a negative control. The main scientific concern is that the authors do not present data to indicate whether autophagy is occurring in host cells that are infected with *Cryptosporidium*. In their experiments, HCT8 cells are infected with parasites which then undergoes multiple cycles of infection and death of the host cell, but not every cell is infected. This makes it difficult to know if they are studying events that relate to the infected cell versus a culture where lots of cells are dying and stressing all cells in the culture. The authors may have the answer to this question in hand from their imaging experiments that can directly address this issue but this is the aspect of the submission that needs the most attention to strengthen the conclusions of this study.

Major Comments:

- Fig 1 A IFITM3 expression - It is not clear what the 3 samples refer to for infected versus uninfected. Are these individual cultures or triplicate samples? This is also an experiment that looks at 24 hours after infection. This allows the parasites to undergo two cycles of replication which then slows down by 48 hours. Since the change in relative expression levels of p62 are first significant at 12 and 24 hours (when cell lysis will be occurring) it would be good to show a time course for IFITM3 expression - does this occur in response to infection or because of cell death in the cultures? It is also unclear from these data sets whether the levels of autophagy are high or low. What would these levels look like compared to cultures treated with compounds or conditions that induce autophagy?
- Figure 2A needs a control to show what the RNAi transfection does to cultures without infection. Without this it is difficult to interpret the data.
- In Fig 2D - it is a challenge to conclude that LC3B levels have been increased but perhaps there are more puncta which could be quantified. The other issue with this figure (and with other parts of the submission) are it is not clear what cells in these images are infected and whether the autophagy events occur in these infected cells. They do have DAPI staining but it is difficult to see the parasites. It is also key, even at this point in the manuscript to know how or if this intervention affects parasite replication. A higher power view of these images may be helpful.
- For the densitometry - what are the error bars derived from - pooled data sets or individual experiments?
- It is not stated in the text, figures, or figure legend how parasite burden is normalized. It appears to be normalized to the "infection" condition at 48hrs as "1", however this needs to be stated. Because the authors normalize the infection condition at 48hrs, the reader does not get a sense for how much parasite growth occurs from time 0 to time 48hr and thus it is difficult to interpret the effects that each condition or treatment has on parasite growth. Each reading at 48hrs should instead be normalized to an input reading at time 0 or within the first few hours of infection. Alternatively, the authors could use a negative control condition that inhibits parasite growth, such as treatment with the drug Nitazoxanide. These important additions will give the reader a better sense for the magnitude of parasite growth and how this is changed upon IFITM3 knockdown or overexpression.
- Figure 5. Details of the time point assessed are not apparent and in the western blot analysis of 5C it looks like cell lysates but these are factors that are typically secreted and can be measured by ELISA. What are the levels of these cytokines in the supernatants?
- Despite the title of their paper, the authors do not show that STAT1 affects parasite growth. To show that the STAT1-IFITM3 pathway is important for defense against *Cryptosporidium*, the authors should measure parasite growth in HCT8s (as in Figure 5) in the presence or absence of the STAT1 inhibitor used in Figure 6.
- Fig 6 - this inhibitor does many things - and in panel 6B the effects on uninfected cells are marked as being significant - which seems at odds with the data presented. The inclusion of internal controls here that assess the impact of the STAT1 inhibitor on uninfected cells indicates that there is a basal STAT1 activity in these cells that drives IFITM3 expression. What is not shown here, that would close the loop, is what is the impact on parasite replication.
- The authors consistently use unpaired T-tests when comparing protein and RNA expression levels across three (or more) experimental conditions. ANOVA tests should be used when comparing more than two groups. The authors mention that ANOVAs were used in the methods section, but no ANOVAs are mentioned in the figure legends.

Minor Comments

- The authors do not title many of their graphs in their figures, especially those presenting qualifications of protein expression. Titles should be included for each graph to better orient the reader.
- The authors include several experiments on how IFITM3 affects cell death and the production of cytokines during infection, however, they do not provide sufficient context for how these processes might promote or impair parasite defense. It is not sufficient to say, "which might be related to the elimination of the parasite" (Line 202). The reasons for why these experiments were performed and a clear interpretation of the results should be included in the results section.
- The authors do not describe how the histogram analyses of Western blots are performed. Do the graphs displaying histogram analyses of protein expression represent a mean of three different western blots that were analyzed? If so, this needs to be more clearly stated. There are instances when the levels of protein represented in the blots do not seem to match the histogram analyses. For example, in Fig 6A, the LC3B ratio seems significantly altered by the STAT1 inhibitor, however, there appears to be no change ("ns") in this condition in the lower right panel of Figure 6B. In the text, the authors state that "autophagy levels decrease". Discrepancies in the data like this should be avoided or explained in the text to avoid confusion/
- The authors should provide quantification of the IF images in figures 2D, 3F, and 4F. It is not sufficient to say that proteins colocalize, that LC3B levels decrease, or that parasite burdens decrease/increase without an actual quantification of the images taken.

Additional comments:

- Line 1: change "immunity" to "cells"
- Line 53: Parasite vacuoles are still intracellular (ie. under/within the cell membrane), however, they can be considered extracytoplasmic
- Line 62: Include importance of interferon lambda in defense and cite papers (Ferguson, 2019 and Gibson, 2022)
- Line 62: Interferon beta has opposing effects, inhibiting cryptosporidium growth in some contexts but also promoting cryptosporidium growth (cite Deng 2023)
- Line 81: change "destroy" to "inhibit" or "block"
- Line 90-93: The study by El-Refai does not make any conclusion about the ATG16L1 polymorphism affects on intestinal inflammation. Nor does this paper mechanistically test how these mutations might affect autophagy in human cells. This

sentence should be reworded to avoid mischaracterization of this paper.

-Line 144: The authors should clearly describe how LC3 and p62 proteins are used as a measurement of cellular autophagy i.e. LC3-II levels typically increase with autophagic activity and p62 levels typically decrease

-Line 148: Autophagic flow was not necessarily blocked, it just did not increase over this time period. This could mean that it was blocked, it paused, or it was not induced

-Line 151: used "decreased" instead of "weakened"

-Line 155: "knock down" instead of "knocked down"

-Line 160: at what time point were these images taken? This needs to be described in the text

-Line 162-163: the author needs to briefly describe the involvement of these proteins in autophagy? Do we expect their decreasing levels to indicate increased or decreased autophagy?

-Line 185: change "co-location" to "colocalization"

-Line 199: the HCT8 immune response should not be considered "intense" as the increases in IL-8 and TNF α are very modest both by western blot and qPCR.

-Line 217: "reduced"

-Line 227: There is a mislabeling of figure panels in the results section for figure 5 so that Fig 5E is not referenced in the text

-Line 265 - autophagy can also have little to no effect on pathogen burden so this should be reworded to remove the word "either"

Figure 1:

-A - The lanes for the western blot should be clearly labeled. It's unclear to the reader what each lane represents. Are these biological replicates, technical replicates, or different time points?

-B - at what time point was RNA collected for qPCR?

-B,C - Include graph titles i.e. "IFITM3 Protein expression"

-E, F - What groups are being compared for the statistical tests in figures E and F. Is each column being compared to the previous column or to the control at 0h? The details of these tests need to be clearly explained in the figure legends.

Figure 2:

-A-C - at what time point after infection were cells harvested for western blots?

-B,C, F - Include graph titles i.e. "IFITM3 Protein expression"

-When comparing more than two experimental groups, authors cannot use unpaired T-test and instead should use ANOVA statistical test

-D - authors should provide some quantitative measure of LC3 puncta and staining from their images

-F - at what time point were these measurements taken?

Figure 3:

-F - the authors should provide quantification for colocalization analysis in figure 3F

-the authors need to indicate the timepoints (hours after infection) for each of these experiments and better describe the data sets shown panels G-I,

Figure 4:

-A, B - when were these measurements taken? Graphs in B need titles.

-Graphs require titles

-F - rigorous quantification of parasite load in IF images is important to determine relative increases and decreases in parasite proliferation across conditions and the time point examined.

-G-H - it is not clear what this data presents. Does "Hs" indicate human DNA? These graphs need titles. While the text says that Cp infection and IFITM3 knockdown does not influence the number of HCT8 cells, there do seem to be significant differences in "Hs" expression across all conditions. Thus, these data contradict what is said in the text.

Figure 5:

-A, B - When are these measurements taken? What marker do the authors use to distinguish early vs. late apoptosis?

-Include graphs of titles

- E - the levels of human "Hs" DNA appear to increase in the Or ifitm3 condition. What affect does overexpression have on cells? What is the overall effect of 3-MA or Or ifitm3 on magnitude of parasite growth normalized to an input time-point?

Figure 6:

-A, B - LC3B levels are clearly down by western blot in the STAT1 inhibitor condition (A) but the quantification (B) shows no significance. Similarly, the STAT1 inhibitor appears to have no affect on p-STAT1 levels in A and B but the graph records a significant difference.

-all graphs need titles

Thank you for your review of our manuscript (Ref No. LSA-2025-03200-T). We appreciate the concerns and suggestions provided by the reviewers, and have revised our manuscript accordingly. Our point-by-point responses are provided below, and text that has been added or modified from the original text is shown in the revised manuscript in red font. We are confident that these changes have both met the reviewers' concerns and have considerably improved the manuscript. We know that your journal has high publication standards, so we have already had the language of this paper corrected by a professional language editing service that specializes in scientific manuscripts.

We wish to thank you for considering this resubmission for publication in your journal.

Sincerely,

Dr. Cao

Reviewer #1 (Comments to the Authors):

Epithelial cell-intrinsic defenses, particularly those activated by interferon signaling, play a crucial role in *Cryptosporidium* clearance, though the underlying regulatory mechanisms are not yet fully understood. IFITM3 is an interferon-stimulated gene that modulates viral infection through autophagy; however, its role in *C. parvum* infection is unknown. This study reports a significant upregulation of IFITM3 expression following *C. parvum* infection in human HCT-8 cells, which may regulate the cell inflammatory response, cell survival, and parasite clearance through the STAT1-IFITM3 axis. In-depth mechanistic details are explored, including the involvement of autophagy. These findings provide new insights into current understanding of how interferon signaling contributes to cell-intrinsic defense to resist *C. parvum* infection, which is relevant to the development of potential therapeutic targets. The experiments were generally well-performed, and the data appear to be

very solid. The manuscript is well-organized and should be of interests to the readers of the journal. My concerns/comments are very minor, as listed below.

1. It would be beneficial to include a discussion on how *C. parvum* infection may induce IFITM3 expression. Specifically, does the infection activate IFN signaling in HCT-8 cells?

Response:

Thank you for your insightful comment. Yes, *Cryptosporidium parvum* infection activates interferon (IFN) signaling in HCT-8 cells, which we believe contributes to the upregulation of IFITM3 expression. This was initially demonstrated in our previous study, published in PLoS Neglected Tropical Diseases (2021, 15(11): e0009949), in which proteomic analysis revealed that the IFN signaling pathway was the most significantly enriched biological process following *C. parvum* infection. Among the 67 significantly upregulated proteins, seven (~10%) were identified as interferon-stimulated genes (ISGs), indicating robust activation of IFN signaling. In the current study, we further validated this by measuring the levels of IFN- α 1 and IFN- β in HCT-8 cells post-infection. Our data (Figure S1B–D) show that both IFN- α 1 and IFN- β protein levels were upregulated in response to infection. These findings support the conclusion that *C. parvum* infection activates IFN signaling, which likely leads to the induction of IFITM3 expression in HCT-8 cells.

2. Cell death through apoptosis could serve as a defense mechanism to clear certain intracellular pathogens. It is particularly interesting that IFITM3 induction can inhibit apoptosis while also enhancing parasite clearance in HCT-8 cells. *C. parvum* infection may induce apoptosis in both directly infected cells and bystander non-infected cells. The observed decrease in infection burden may simply resulted from the increased survival of bystander non-infected cells. The methods used to measure and calculate the infection burden should be clarified and discussed.

Response:

Thank you for your constructive suggestions. We have clarified and elaborated on the methodology used to quantify *C. parvum* infection burden in the "Calculation of cell numbers and *C. parvum* growth" section of the Methods (Lines 519–529). The proliferation of *C. parvum* oocysts was quantified using SYBR Green-based qPCR targeting Cp-18S (parasite-specific) and Hs-18S (host-specific) transcripts. Standard curves were established by infecting uninfected HCT-8 cells with serially diluted oocyst suspensions (1×10^1 to 1×10^5 oocysts/sample), followed by DNA extraction and qPCR. Δ Ct values (Cp18S - Hs18S) were plotted against the log of oocyst input to generate a standard curve, allowing for both absolute quantification and relative parasite load assessment (see Figure S2). To account for potential confounding effects

due to the enhanced survival of bystander non-infected cells, we now primarily rely on absolute quantification of parasite burden based on Ct values and the standard curve. This method more accurately reflects actual oocyst counts, independent of host cell viability or density. This qPCR-based approach was adapted from the protocol described in: "High-Throughput Screening of Drugs Against the Growth of *C. parvum* In Vitro by qRT-PCR" (Methods Mol Biol. 2020).

3. Some paragraphs in the discussion section are quite lengthy and resemble a description of the results rather than a discussion. These should be shorted for clarity and focus.

Response:

Thank you for your valuable feedback. In response to your suggestion, we have carefully revised the Discussion section by streamlining several lengthy paragraphs and reducing redundancy. We have ensured that the revised text emphasizes interpretation, significance, and context of our findings, rather than repeating detailed results.

4. There are several typos in the manuscripts, such as TNF-a (rather than TNF- α) in Figures.

Response:

Thank you for pointing this out. We have carefully reviewed the manuscript, including all figure labels and legends, and corrected typographical errors such as "TNF-a" to the proper format "TNF- α ". We have also checked for and corrected similar issues throughout the manuscript to ensure consistency and accuracy in the use of Greek symbols and scientific terminology.

Reviewer #2 (Comments to the Authors):

In their manuscript title "STAT1-IFITM3 promotes autophagy in epithelial immunity to control *Cryptosporidium parvum* infection", Cui et al. build on recent work in the immunity to *Cryptosporidium* field to show that IFITM3, an interferon stimulated gene important for anti-viral defense and induction of autophagy, can promote control of *Cryptosporidium* in the human colonic adenocarcinoma HCT-8 cell line and is induced downstream of STAT1. This work has the potential to increase our understanding of how autophagy is induced in cultures infected with *Cryptosporidium* and suggest a novel role for the interferon stimulated gene IFITM3 in restriction of parasite growth.

In principle, these studies are of interest to the *Cryptosporidium* community, but

there are questions about how the data are presented that make it difficult to assess the manuscript. Below is an extensive list of question but it is hoped that most of these can be addressed with the data sets in hand and not necessarily require new experiments. A theme for some of these is the need for more rigor to measure parasite burden relative to an input value or a negative control. The main scientific concern is that the authors do not present data to indicate whether autophagy is occurring in host cells that are infected with *Cryptosporidium*. In their experiments, HCT8 cells are infected with parasites which then undergoes multiple cycles of infection and death of the host cell, but not every cell is infected. This makes it difficult to know if they are studying events that relate to the infected cell versus a culture where lots of cells are dying and stressing all cells in the culture. The authors may have the answer to this question in hand from their imaging experiments that can directly address this issue but this is the aspect of the submission that needs the most attention to strengthen the conclusions of this study.

Major Comments:

- Fig 1 A IFITM3 expression - It is not clear what the 3 samples refer to for infected versus uninfected. Are these individual cultures or triplicate samples? This is also an experiment that looks at 24 hours after infection. This allows the parasites to undergo two cycles of replication which then slows down by 48 hours. Since the change in relative expression levels of p62 are first significant at 12 and 24 hours (when cell lysis will be occurring) it would be good to show a time course for IFITM3 expression - does this occur in response to infection or because of cell death in the cultures? It is also unclear from these data sets whether the levels of autophagy are high or low. What would these levels look like compared to cultures treated with compounds or conditions that induce autophagy?

Response:

Thank you for your valuable comments. We have addressed your concerns as follows: 1) The three samples shown in Figure 1A represent three independent biological replicates, i.e., three individually cultured wells for both infected and uninfected groups at 24 hours post-infection (hpi). 2) As you suggested, we have replaced the original 24 h-only data with a time course of IFITM3 expression (now shown in the revised Figure 1A), which spans from 0 to 48 hpi. This helps better evaluate the dynamic response of IFITM3 to *C. parvum* infection. 3) As referenced in the article by Love et al. (2019, Nat Commun 10:1862), *C. parvum* completes multiple rounds of asexual replication by 36 hpi, and sexual differentiation typically begins around 42 hpi. At 24 hpi, parasite egress and early stages of host cell lysis may occur, but most

HCT-8 cells remain viable, with only mild apoptosis observed (see Figure S3). Therefore, the observed upregulation of IFITM3 and p62 at 12–24 hpi is most likely a direct response to infection, rather than a secondary effect of widespread cell death. Moreover, prior to sample collection, we carefully removed dead cells and debris from the culture supernatant to minimize their impact on the analysis. 4) The transient increase in p62 expression at 12–24 hpi may reflect impaired autophagic flux due to disrupted autophagosome–lysosome fusion, which is known to occur in response to intracellular pathogens. For example, *Mycobacterium tuberculosis* can block autophagosome maturation through the SecA2 pathway (Clemens et al., PLoS Pathog, 2018). We hypothesize that a similar mechanism might explain p62 accumulation at this stage of *C. parvum* infection, and we plan to explore this further in future studies. 5) To help contextualize the level of autophagy induced by *C. parvum*, we included experiments using Rapamycin, a known autophagy activator. The expression levels of LC3B-II/I and p62 in Rapamycin-treated cultures are shown in Figure 1E–G, and provide a benchmark against which infection-induced autophagy can be compared. 6) As shown in Figure 2, knockdown of IFITM3 resulted in decreased LC3B-II/LC3B-I ratios and increased p62 accumulation in infected cells, suggesting that IFITM3 promotes autophagic activity during *C. parvum* infection. This supports the conclusion that IFITM3 contributes to host defense by enhancing autophagic responses.

- Figure 2A needs a control to show what the RNAi transfection does to cultures without infection. Without this it is difficult to interpret the data.

Response:

Thank you for your suggestion. To address this concern, we have added an appropriate control group to Figure 2A–L: the *siIFITM3* (uninfected) group. This control reflects the effect of RNAi transfection on HCT-8 cells in the absence of *C. parvum* infection. Including this group allows for direct comparison between infected and uninfected conditions, and helps clarify whether the observed changes are infection-specific or due to IFITM3 knockdown alone.

- In Fig 2D - it is a challenge to conclude that LC3B levels have been increased but perhaps there are more puncta which could be quantified. The other issue with this figure (and with other parts of the submission) are it is not clear what cells in these images are infected and whether the autophagy events occur in these infected cells. They do have DAPI staining but it is difficult to see the parasites. It is also key, even at this point in the manuscript to know how or if this intervention affects parasite replication. A higher power view of these images may be helpful.

Response:

Thank you for your valuable and insightful suggestions. We have addressed your concerns through the following revisions and clarifications: 1) To better support our interpretation of LC3B expression changes, we quantified the number of LC3B puncta per cell using fluorescence microscopy. The quantitative results are now shown in Figure 2E, providing clear evidence that LC3B puncta increased significantly following *C. parvum* infection, and were reduced upon IFITM3 knockdown. 2) To determine whether autophagy occurs specifically in infected cells, we performed dual immunofluorescence staining for both *C. parvum* (green) and LC3B (red). The merged images (Figure 2K) and quantification (Figure 2L) indicate that autophagy occurs in both infected and uninfected cells, but is more pronounced in infected cells. These results suggest that infection directly enhances autophagy in host cells. 3) To assess whether soluble factors secreted from *C. parvum* and infected cells contribute to autophagy induction, we treated HCT-8 cells with culture supernatants from infected or uninfected cells (24 hpi). We observed a slight increase in IFITM3 and LC3B expression in cells exposed to supernatants from infected cultures (Figure 2M and N), suggesting that secreted factors—such as exosomes or cytokines like IFNs—may also induce autophagy in neighboring bystander cells. 4) It is also possible that some cells initially infected by *C. parvum* have already released parasites by the time of observation, but remain alive and continue to exhibit autophagic activity. This may partially explain the presence of autophagy markers in cells that appear uninfected. 5) We agree that higher resolution imaging enhances interpretation. We have now included higher power views in the revised version of Figure 2K to better distinguish parasite localization and autophagic markers within individual cells.

- For the densitometry - what are the error bars derived from - pooled data sets or individual experiments?

Response:

Thank you for your thoughtful question. The error bars presented in all densitometry graphs represent the standard deviation (SD) calculated from at least three independent biological experiments, each performed separately. This information has now been explicitly added to the “Statistical analysis” section of the Methods and Materials (Line 569) to ensure clarity and transparency.

- It is not stated in the text, figures, or figure legend how parasite burden is normalized. It appears to be normalized to the "infection" condition at 48hrs as "1", however this needs to be stated. Because the authors normalize the infection condition at 48hrs, the reader does not get a sense for how much parasite growth occurs from time 0 to time 48hr and thus it is difficult to interpret the effects that each condition or

treatment has on parasite growth. Each reading at 48hrs should instead be normalized to an input reading at time 0 or within the first few hours of infection. Alternatively, the authors could use a negative control condition that inhibits parasite growth, such as treatment with the drug Nitazoxanide. These important additions will give the reader a better sense for the magnitude of parasite growth and how this is changed upon IFITM3 knockdown or overexpression.

Response:

Thank you for your thoughtful and constructive suggestions. We agree that normalization based on initial parasite input or the use of a defined inhibitory control condition would improve interpretability. The parasite burden was measured using SYBR Green-based qPCR targeting *C. parvum* 18S rRNA (Cp-18S) and normalized to host 18S rRNA (Hs-18S) as an internal control. A standard curve was generated using serially diluted oocyst input (1×10^1 to 1×10^5 oocysts/sample), and ΔC_T values were plotted against log-transformed oocyst numbers. The absolute oocyst counts were calculated based on this standard curve, allowing for quantification of parasite growth independent of internal normalization assumptions (see Figure S2). This methodology is detailed in the “Calculation of cell numbers and *C. parvum* growth” section (Lines 519–529) of the Methods and is based on a previously published protocol: High-Throughput Screening of Drugs Against the Growth of *C. parvum* In Vitro by qRT-PCR (Methods Mol Biol. 2020). To better represent magnitude of parasite proliferation, we now report absolute oocyst counts rather than normalizing all data to the 24 hpi “infection” condition. This allows for a clearer view of how parasite numbers change over time and under different treatment conditions.

We also appreciate the suggestion of including a reference control such as Nitazoxanide. While this was not included in the current set of experiments, we agree it would serve as a valuable benchmark for evaluating treatment efficacy and plan to incorporate it into future work. In the current study, parasite burden in IFITM3 knockdown or overexpression groups was compared directly with the infected control group at the same timepoint, with consistent input parasite loads and cell numbers across groups. We have revised the relevant figure legends and text to clarify our normalization approach and use of absolute quantification where applicable.

- Figure 5. Details of the time point assessed are not apparent and in the western blot analysis of 5C it looks like cell lysates but these are factors that are typically secreted and can be measured by ELISA. What are the levels of these cytokines in the supernatants?

Response:

Thank you for your helpful and insightful comments. We have clarified in the figure legend and relevant text that all measurements in Figure 5 were conducted at 24 hours

post-infection (hpi), unless otherwise stated. We agree that IL-8 and TNF- α are secreted cytokines and that their levels are more appropriately assessed in the culture supernatants rather than in cell lysates. To address this, we conducted additional experiments using enzyme-linked immunosorbent assay (ELISA) to quantify cytokine levels in the supernatants of each group. As now shown in Figure 5H and 5I, inhibition of autophagy using 3-MA led to reduced secretion of IL-8, while overexpression of IFITM3 significantly increased the levels of both IL-8 and TNF- α in the supernatants. Similarly, Figure 4J and 4K show ELISA results from earlier experiments, where IL-8 and TNF- α were elevated in the infected group and notably decreased in the siIFITM3-infected group, especially IL-8. We have added a detailed description of the ELISA procedure to the “Methods and Materials” section under the heading "Enzyme-linked immunosorbent assay (ELISA)" (Lines 544–548).

- Despite the title of their paper, the authors do not show that STAT1 affects parasite growth. To show that the STAT1-IFITM3 pathway is important for defense against *Cryptosporidium*, the authors should measure parasite growth in HCT8s (as in Figure 5) in the presence or absence of the STAT1 inhibitor used in Figure 6.

Response:

Thank you for this important observation. In response to your suggestion, we conducted additional experiments to evaluate parasite growth in HCT-8 cells treated with or without the STAT1 inhibitor, using the same qPCR-based method described previously. The results are now presented in Figure 6H and 6I. We found that inhibition of STAT1, in the context of *C. parvum* infection, did not affect the number of HCT-8 host cells, but significantly increased the parasite burden. This supports our hypothesis that STAT1 contributes to host defense by limiting *C. parvum* replication.

- Fig 6 - this inhibitor does many things - and in panel 6B the effects on uninfected cells are marked as being significant - which seems at odds with the data presented. The inclusion of internal controls here that assess the impact of the STAT1 inhibitor on uninfected cells indicates that there is a basal STAT1 activity in these cells that drives IFITM3 expression. What is not shown here, that would close the loop, is what is the impact on parasite replication.

Response:

Thank you for your thoughtful and detailed feedback. We appreciate you pointing out this inconsistency. You are correct that the previously presented data in Figure 6B was inaccurate due to a mistake during figure preparation. We have now corrected this error in the revised figure to ensure consistency with the actual data. The updated analysis reflects that the effect of the STAT1 inhibitor on uninfected cells is modest, though still detectable, suggesting a basal level of STAT1 activity that contributes to

IFITM3 expression even in the absence of infection. As you rightly noted, the functional impact of STAT1 inhibition on *C. parvum* replication is critical to support the role of the STAT1–IFITM3 pathway in host defense. To address this, we measured parasite burden in HCT-8 cells with or without STAT1 inhibitor treatment. The results are now shown in Figure 6H and 6I.

- The authors consistently use unpaired T-tests when comparing protein and RNA expression levels across three (or more) experimental conditions. ANOVA tests should be used when comparing more than two groups. The authors mention that ANOVAs were used in the methods section, but no ANOVAs are mentioned in the figure legends.

Response:

Thank you for pointing this out. We apologize for the oversight in the figure legends. As correctly noted, unpaired Student's t-tests were used for comparisons between two groups, while one-way analysis of variance (ANOVA) was applied for comparisons involving three or more groups, as stated in the Statistical Analysis section of the Methods. We have now revised all relevant figure legends to clearly indicate when ANOVA was used, ensuring consistency between the figure legends and the statistical methods described in the text.

Minor Comments

- The authors do not title many of their graphs in their figures, especially those presenting qualifications of protein expression. Titles should be included for each graph to better orient the reader.

Response:

Thank you for your helpful suggestion. In response, we have carefully reviewed all figures and have now added titles to each graph. These titles are intended to clearly indicate the content and purpose of each graph, thereby improving readability and helping readers more easily interpret the data.

- The authors include several experiments on how IFITM3 affects cell death and the production of cytokines during infection, however, they do not provide sufficient context for how these processes might promote or impair parasite defense. It is not sufficient to say, "which might be related to the elimination of the parasite" (Line 202). The reasons for why these experiments were performed and a clear interpretation of the results should be included in the results section.

Response:

Thank you for your valuable suggestion. In response, we have revised the Results section (Lines 209 and 217) to provide a more detailed explanation of the rationale for examining cell death and cytokine production in the context of *C. parvum* infection. We updated the Discussion section to better integrate these findings into the broader context of epithelial cell-intrinsic immunity. These revisions provide a more coherent interpretation of the functional role of IFITM3 in host defense, beyond simple correlative observations.

- The authors do not describe how the histogram analyses of Western blots are performed. Do the graphs displaying histogram analyses of protein expression represent a mean of three different western blots that were analyzed? If so, this needs to be more clearly stated. There are instances when the levels of protein represented in the blots do not seem to match the histogram analyses. For example, in Fig 6A, the LC3B ratio seems significantly altered by the STAT1 inhibitor, however, there appears to be no change ("ns") in this condition in the lower right panel of Figure 6B. In the text, the authors state that "autophagy levels decrease". Discrepancies in the data like this should be avoided or explained in the text to avoid confusion/

Response:

Thank you for your helpful and detailed observations. We have now clarified the methodology for histogram analyses in the “Western blotting” section of the Methods and Materials (Lines 479–481). Specifically, all histogram data represent the mean \pm SD of densitometric analyses from at least three independent Western blot experiments, each normalized to the internal control. Regarding Figure 6A and 6B, we agree that there was a discrepancy between the blot image and the quantification panel. Upon review, we identified an inconsistency in the data processing and have since repeated the experiment, followed by re-analysis and quantification. The updated results now more accurately reflect that autophagy levels, as measured by LC3B-II/LC3B-I ratio, are decreased in the STAT1-inhibited group compared with the untreated control, as shown in the revised Figure 6E. We have also updated the corresponding figure legend and results description to ensure that the visual and quantitative data are in full agreement and clearly interpreted, avoiding potential confusion for readers.

- The authors should provide quantification of the IF images in figures 2D, 3F, and 4F. It is not sufficient to say that proteins colocalize, that LC3B levels decrease, or that parasite burdens decrease/increase without an actual quantification of the images taken.

Response:

Thank you for your helpful suggestion. In response, we have now provided quantitative analyses of the immunofluorescence (IF) images. These quantifications strengthen the conclusions drawn from the microscopy data and ensure objective measurement rather than descriptive interpretation alone.

Additional comments:

-Line 1: change "immunity" to "cells"

Response:

Thank you for your suggestion. We have revised Line 1 by changing the word “immunity” to “cells” as requested.

-Line 53: Parasite vacuoles are still intracellular (ie. under/within the cell membrane), however, they can be considered extracytoplasmic

Response:

Thank you for your helpful clarification. In response, we have revised the wording in Line 54, changing “extracellular” to “extracytoplasmic” to more accurately describe the location of the parasite vacuoles.

-Line 62: Include importance of interferon lambda in defense and cite papers (Ferguson, 2019 and Gibson, 2022)

Response:

Thank you for your helpful suggestion. We have revised the text in Line 64–67 to include the importance of interferon lambda (IFN- λ) in host defense against *Cryptosporidium*, and we have cited the recommended references: Ferguson et al., 2019 and Gibson et al., 2022.

-Line 62: Interferon beta has opposing effects, inhibiting cryptosporidium growth in some contexts but also promoting cryptosporidium growth (cite Deng 2023)

Response:

Thank you for your suggestion. We have revised the text in Line 62–64 to state that interferon beta (IFN- β) exhibits opposing effects, including both inhibitory and promotive roles in *Cryptosporidium* infection depending on the context, and have cited the recommended study (Deng, 2023) to support this observation.

-Line 87: change "destroy" to "inhibit" or "block"

Response:

Thank you for your suggestion. We have revised the wording in Line 89, changing “destroy” to “inhibit” accordingly.

-Line 90-93: The study by El-Refai does not make any conclusion about the ATG16L1 polymorphism affects on intestinal inflammation. Nor does this paper mechanistically test how these mutations might affect autophagy in human cells. This sentence should be reworded to avoid mischaracterization of this paper.

Response:

Thank you for your helpful suggestion. Upon re-evaluating the study by El-Refai et al., we acknowledge that while the authors state in their conclusion that the ATG16L1 SNP may impair autophagy and increase susceptibility to *C. parvum* infection, they do not directly examine the impact of this polymorphism on intestinal inflammation, nor do they mechanistically test its effect on autophagy in human cells. To ensure accurate representation of their findings, we have reworded the sentence in Line 96 to: “ An autophagy-related 16-like 1 (ATG16L1) gene polymorphism has been associated with increased risk and severity of *C. parvum* infection.” To further strengthen this section, we have added a recent mechanistic study by Sharmin et al., 2024, highlighting the role of long non-coding RNA in IFN- γ -mediated epithelial defense: “ A recent study demonstrated that the lncRNA Nostrill enhances transcription of IFN- γ -induced genes such as *Igtp*, *Gadd45g*, and *iNos*, thereby positively regulating autophagy and promoting epithelial cell-intrinsic defense against *Cryptosporidium* infection (Sharmin, 2024)” (Lines 98–102).

-Line 144: The authors should clearly describe how LC3 and p62 proteins are used as a measurement of cellular autophagy i.e. LC3-II levels typically increase with autophagic activity and p62 levels typically decrease

Response:

Thank you for your suggestion. To improve clarity, we have added the following explanatory sentence to Line 150: “Typically, LC3B-II/LC3B-I ratios increase with autophagic activity, while p62 levels decrease, as p62 is degraded during autophagic flux.”

-Line 148: Autophagic flow was not necessarily blocked, it just did not increase over this time period. This could mean that it was blocked, it paused, or it was not induced

Response:

Thank you for your insightful comment. To avoid overinterpretation, we have revised the statement in Line 159 to more accurately reflect the possibilities, now reading:

“These findings suggest that autophagic flux may have been blocked, paused, or not induced at 12 and 24 hours post-infection.”

-Line 151: used "decreased" instead of "weakened"

Response:

Thank you for your suggestion. We have revised the wording in Line 161, replacing “weakened” with “decreased” to ensure more accurate and scientific language.

-Line 155: "knock down" instead of "knocked down"

Response:

Thank you for your suggestion. We have revised the wording in Line 165, changing “knocked down” to “knock down” for grammatical accordingly.

-Line 160: at what time point were these images taken? This needs to be described in the text

Response:

Thank you for your suggestion. The images were taken at 24 hours post-infection (hpi), and we have now added this information to Line 172 to clearly indicate the time point of image acquisition.

-Line 162-163: the author needs to briefly describe the involvement of these proteins in autophagy? Do we expect their decreasing levels to indicate increased or decreased autophagy?

Response:

Thank you for your suggestion. We have added a brief explanation in Lines 173–175 to clarify the roles of these proteins. We now state: “Beclin-1, ATG7, and ATG5 are essential components of the autophagy machinery, involved in autophagosome formation and maturation. Their expression is typically elevated during active autophagy; therefore, decreased levels of these proteins suggest a reduction in autophagic activity.”

-Line 185: change "co-location" to "colocalization"

Response:

Thank you for your suggestion. We have corrected the terminology in Line 203, changing “co-location” to “colocalization” accordingly.

-Line 199: the HCT8 immune response should not be considered "intense" as the increases in IL-8 and TNFa are very modest both by western blot and qPCR.

Response:

Thank you for your suggestion. To more accurately reflect the data, we have deleted the term “intense” in Line 219 when describing the HCT-8 immune response.

-Line 217: "reduced"

Response:

Thank you for your suggestion. We have revised the wording in Line 238, replacing the original term with “reduced” for greater clarity and precision.

-Line 227: There is a mislabeling of figure panels in the results section for figure 5 so that Fig 5E is not referenced in the text

Response:

Thank you for pointing this out. We have carefully reviewed the figure references and corrected the mislabeling in Line 246, ensuring that Figure 5E is now properly referenced in the text and that all panel labels are consistent with the figure layout.

-Line 265 - autophagy can also have little to no effect on pathogen burden so this should be reworded to remove the word "either"

Response:

Thank you for your suggestion. To more accurately reflect the range of autophagy outcomes, we have replaced the word “either” with “could” in Line 289, acknowledging that autophagy may increase, decrease, or have minimal impact on pathogen burden depending on the context.

Figure 1:

-A - The lanes for the western blot should be clearly labeled. It's unclear to the reader what each lane represents. Are these biological replicates, technical replicates, or different time points?

-B - at what time point was RNA collected for qPCR?

-B,C - Include graph titles i.e. "IFITM3 Protein expression"

-E, F - What groups are being compared for the statistical tests in figures E and F. Is each column being compared to the previous column or to the control at 0h? The details of these tests need to be clearly explained in the figure legends.

Response:

Thank you for your thoughtful and constructive suggestions. We have addressed each point as follows: 1) Western blot lane clarification in Figure 1A: The three lanes represent three independent biological replicates (i.e., individual cultures) for both infected and uninfected HCT-8 cells at 24 hours post-infection (hpi). However, based on your feedback, we have replaced this panel with a time-course analysis of IFITM3 protein expression, now shown in the updated Figure 1A, to provide more meaningful insights. 2) RNA collection time point in Figure 1B: The RNA used for qPCR analysis was collected at 24 hpi, and this information has now been added to the

figure legend. The corresponding data is shown in Supplementary Figure 1A. 3) Graph titles in Figures 1B and 1C: Titles such as “Relative protein expression level of IFITM3” are now clearly labeled in the Y-axis titles of the histogram plots for clarity. We have also ensured consistency in graph titles throughout the manuscript. 4) Statistical comparisons in Figures 1E and 1F: In Figures 1E and 1F, each experimental time point is compared to the 0-hour control. This has now been explicitly stated in the figure legend for Figure 1 (Line 848) to avoid ambiguity and ensure accurate interpretation of the statistical analysis.

Figure 2:

- A-C - at what time point after infection were cells harvested for western blots?
- B,C, F - Include graph titles i.e. "IFITM3 Protein expression"
- When comparing more than two experimental groups, authors cannot use unpaired T-test and instead should use ANOVA statistical test
- D - authors should provide some quantitative measure of LC3 puncta and staining from their images
- F - at what time point were these measurements taken?

Response:

Thank you for your valuable suggestions. We have made the following revisions to address your points: 1) The cells were harvested at 24 hours post-infection (hpi) for western blot analysis, and we have added this information in Line 866 for clarity. All experiments from this point onward were performed at 24 hpi. 2) We have added descriptive titles to the graphs, such as “Relative protein expression level of IFITM3”, which now appear in the Y-axis titles of the histograms for better clarity and interpretation. 3) As per your suggestion, we used ANOVA to compare more than two experimental groups. However, for comparisons between specific groups, we used unpaired t-tests to ensure appropriate statistical analysis. We have clarified this in the figure legend to specify how comparisons were made between each group. 4) We have included quantitative data on LC3 puncta and staining in Figures 2F and 2L, which now show the number of puncta per cell and provide more detailed data to support our conclusions. 5) The measurements in Figure 2F were taken at 24 hpi, and we have added this information in Lines 867 and 868 to ensure clarity regarding the timing of these measurements.

Figure 3:

- F - the authors should provide quantification for colocalization analysis in figure 3F
- the authors need to indicate the timepoints (hours after infection) for each of these experiments and better describe the data sets shown panels G-I,

Response:

Thank you for your helpful suggestions. We have addressed your comments as follows:

1) We have added quantitative analysis of colocalization in Figure 3I, which measures the degree of overlap between the relevant fluorescence signals. This provides a more objective and robust assessment of the colocalization observed in Figure 3F. 2) All experiments in Figure 3 were conducted at 24 hpi. We have now clearly stated this in Lines 885–888 of the manuscript. 3) We have expanded the description of the data sets shown in Figures 3K–N (previously labeled G–I) in the Results section (Lines 205–207) to clarify the experimental design, the variables measured, and the relevance of the findings to STAT1 and IFITM3 function in *C. parvum* infection.

Figure 4:

-A, B - when were these measurements taken? Graphs in B need titles.

-Graphs require titles

-F - rigorous quantification of parasite load in IF images is important to determine relative increases and decreases in parasite proliferation across conditions and the time point examined.

-G-H - it is not clear what this data presents. Does "Hs" indicate human DNA? These graphs need titles. While the text says that Cp infection and IFITM3 knockdown does not influence the number of HCT8 cells, there do seem to be significant differences in "Hs" expression across all conditions. Thus, these data contradict what is said in the text.

Response:

Thank you for your detailed and helpful suggestions. We have addressed each point as follows: 1) All measurements presented in Figure 4 were taken at 24 hpi. This information has now been added to the figure legend and clarified in the main text at Line 903) We have added clear titles to all graphs, such as “Relative expression of IFITM3” and “Quantification of parasite burden,” which now appear in the Y-axis labels and/or headers for better readability and interpretation. 3) We agree that accurate quantification is essential. To address this, we have provided a rigorous quantification of *C. parvum* load from immunofluorescence (IF) images in Figure 4M, which now includes statistical analysis to support the observed trends in parasite proliferation. 4) “Hs” refers to host (human) 18S rRNA, used as an internal control for normalization in qPCR. We acknowledge that the original description was misleading. Upon re-analysis, we found variability in host 18S levels across conditions. As such, we have corrected the statement in the text and updated Figure 4N to more accurately reflect the observed differences in HCT-8 cell numbers.

Figure 5:

-A, B - When are these measurements taken? What marker do the authors use to distinguish early vs. late apoptosis?

-Include graphs of titles

- E - the levels of human "Hs" DNA appear to increase in the Or ifitm3 condition. What affect does overexpression have on cells? What is the overall effect of 3-MA or Or ifitm3 on magnitude of parasite growth normalized to an input time-point?

Response:

Thank you for your thoughtful suggestions. We have addressed each of your points as follows: 1) All measurements in Figure 5 were performed at 24 hpi, and this information has now been added to the text in Line 919. To distinguish between early and late apoptosis, we used Annexin V/7-AAD and Annexin V/PI staining: Annexin V⁺/7-AAD⁻ or Annexin V⁺/PI⁻ indicates early apoptosis. Annexin V⁺/7-AAD⁺ or Annexin V⁺/PI⁺ indicates late apoptosis. This clarification has also been added to the Methods section in Line 516. 2) We have added descriptive titles to all graphs, including Y-axis labels to improve clarity and guide interpretation. 3) To address the observation of increased Hs-18S expression in the IFITM3 overexpression group, we repeated the qPCR experiments measuring both Hs-18S (host) and Cp-18S (parasite) transcripts. The updated results are now presented in Figure 5J and 5K. These data confirm that neither *C. parvum* infection, 3-MA treatment, nor IFITM3 overexpression significantly affects HCT-8 cell numbers. We now rely on absolute oocyst counts to evaluate parasite burden more accurately. Using this method, we found that: 3-MA treatment led to a marked increase in *C. parvum* replication. *IFITM3* overexpression resulted in a significant reduction in parasite burden

Figure 6:

-A, B - LC3B levels are clearly down by western blot in the STAT1 inhibitor condition (A) but the quantification (B) shows no significance. Similarly, the STAT1 inhibitor appears to have no affect on p-STAT1 levels in A and B but the graph records a significant difference.

-all graphs need titles

Response:

Thank you for your insightful observations. We have addressed the discrepancies and suggestions as follows: 1) We appreciate your comment regarding inconsistencies between the western blot images and the corresponding quantification. To address this, we repeated the experiment and reanalyzed the densitometry data for both p-STAT1 and LC3B. The updated results are now presented in Figure 6B and 6E. The new data more accurately reflect that both p-STAT1 levels and autophagy activity (as indicated

by LC3B-II/LC3B-I ratio) are significantly reduced in the STAT1-inhibited group compared to the untreated control, aligning with the visual interpretation of the blots in Panel A. 2) We have added descriptive titles to all graphs accordingly, which are now clearly indicated in the Y-axis labels and graph.

June 16, 2025

RE: Life Science Alliance Manuscript #LSA-2025-03200-TR

Prof. Jianping Cao
National Institute for Parasitic Diseases
Key Laboratory of Parasite and Vector Biology
No. 207, Rui Jin 2nd Road
Shanghai 200025
China

Dear Dr. Cao,

Thank you for submitting your revised manuscript entitled "STAT1-IFITM3 promotes autophagy in epithelial immunity to control *Cryptosporidium parvum* infection". As you will see, both reviewers now recommend publication of this manuscript. We would be happy to publish your paper in Life Science Alliance pending final revisions necessary to meet our formatting guidelines.

- Please be sure that the authorship listing and order is correct.
- Please add ORCID ID for secondary corresponding author -- they should have received instructions on how to do so.
- Please add the X and Bluesky handles of your host institute/organization, as well as your own and/or one of the authors, in our system.
- The titles in the system and the manuscript file must be consistent.
- We encourage you to revise the figure legend for Figure 6 such that the figure panels are introduced in alphabetical order.
- Since figure S2 has only one panel, it is unnecessary to label it as A. Please correct the figure and call-out in the manuscript text.
- Please add callouts for Figures S1A-D and S3A-B to your main manuscript text.

A. FINAL FILES:

B. MANUSCRIPT ORGANIZATION AND FORMATTING:

Sincerely,

Reviewer #1 (Comments to the Authors (Required)):

The authors did an outstanding job revising the manuscript and have addressed all of my previous concerns.

Reviewer #2 (Comments to the Authors (Required)):

The initial submission was of general interest but an extensive list of questions was provided that required clarification and/or additional experiments. The authors have extensively revised the manuscript and addressed all questions raised in a thoughtful fashion. We think this is a useful study that many in the field (beyond just *Cryptosporidium*) will find of interest.

June 25, 2025

RE: Life Science Alliance Manuscript #LSA-2025-03200-TRR

Prof. Jianping Cao
National Institute for Parasitic Diseases
Key Laboratory of Parasite and Vector Biology
No. 207, Rui Jin 2nd Road
Shanghai 200025
China

Dear Dr. Cao,

Thank you for submitting your Research Article entitled "STAT1-IFITM3 promotes autophagy in epithelial cells to control *Cryptosporidium parvum* infection". It is a pleasure to let you know that your manuscript is now accepted for publication in Life Science Alliance. Congratulations on this interesting work.

DISTRIBUTION OF MATERIALS:

Again, congratulations on a very nice paper. I hope you found the review process to be constructive and are pleased with how the manuscript was handled editorially. We look forward to future exciting submissions from your lab.

Sincerely,
